# LogSTOP: Temporal Scores over Prediction Sequences for Matching and Retrieval

## Abstract

Neural models such as YOLO and HuBERT can be used to detect local properties such as objects ("car") and emotions ("angry") in individual frames of videos and audio clips respectively. The likelihood of these detections is indicated by scores in [0, 1]. Lifting these scores to temporal properties over sequences can be useful for several downstream applications such as query matching (e.g., "does the speaker eventually sound happy in this audio clip?"), and ranked retrieval (e.g., "retrieve top 5 videos with a 10 second scene where a car is detected until a pedestrian is detected"). In this work, we formalize this problem of assigning Scores for TempOral Properties (STOPs) over sequences, given potentially noisy score predictors for local properties. We then propose a scoring function called LogSTOP that can efficiently compute these scores for temporal properties represented in Linear Temporal Logic. Empirically, LogSTOP, with YOLO and HuBERT, outperforms Large Vision / Audio Language Models and other Temporal Logic-based baselines by at least 16% on query matching with temporal properties over objects-in-videos and emotions-in-speech respectively. Similarly, on ranked retrieval with temporal properties over objects and actions in videos, LogSTOP with Grounding DINO and SlowR50 reports at least a 19% and 16% increase in mean average precision and recall over zero-shot text-to-video retrieval baselines respectively.

## 1 Introduction

Detecting complex temporal events in unstructured data sequences such as videos and audio clips is important in several domains. For instance, traffic surveillance systems need to *match* scenes perceived by autonomous vehicles against critical *temporal* properties such as "the vehicle *always* remains in a given lane". Similarly, search engines might need to *rank* videos or audio clips by relevance to temporal scenes ("a 10 second scene where a person *eventually* starts running" or "a 20-30 second segment where speaker A sounds sad and B sounds frustrated *until* both sound neutral").

Recent work on temporal event detection in videos (Yang et al., 2023; Choi et al., 2024) has focused on using neural detection models such as YOLO (Redmon et al., 2016) to detect objects (for example, "car") in individual video frames, and employing off-the-shelf model checkers such as STORM (Hensel et al., 2022) to verify if the sequence of detection scores satisfies a temporal property. Inspired by these works, we introduce the problem of lifting scores for local properties to *Scores for TempOral Properties* (STOPs) over sequences. Concretely,

*Given a temporal property and (potentially noisy) predictors for local properties, how can we assign a score for a sequence expressing the temporal property?*

The sequences and local properties could correspond to arbitrary modalities and classes of interest, including objects or actions in videos, and speakers or emotions in audio clips. These STOPs are useful for several downstream applications such as **query matching** , i.e., checking if the scores are over a threshold to decide if the sequence expresses a temporal property, and **ranked retrieval**, i.e., ranking sequences against a temporal query by these scores to provide the top-k most relevant results.

We argue that Linear Temporal Logic, with temporal operators such as "Always" ($\square$) and "Until" ($\mathcal{U}$), provides a suitable language for expressing diverse temporal properties of interest. For example, the property "A and B sound happy until A sounds sad" can be written in LTL as $(\text{happy}_A \wedge \text{happy}_B) \, \mathcal{U}$

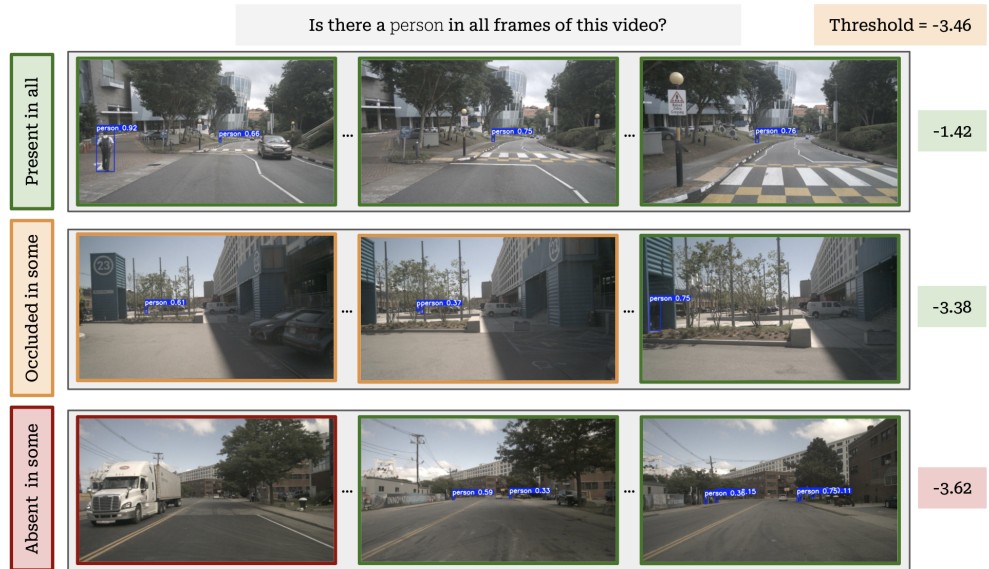

Figure 1: LogSTOPs for three videos with respect to the query *"Is there a person in all frames of this video?"*. Video 2 with occluded persons is assigned a lower score than video 1 (where a person is visible in all frames), and higher score than video 3 (where there are frames with no persons). These scores can be used for ranking and query matching with the adaptive threshold we define in Section 3.1. YOLOv8x is used here to detect objects in individual frames of the videos.

$sad_A$. For temporal properties in LTL, the framework proposed by Yang et al. (2023) can be used as a solution to the STOP problem with two major caveats. Firstly, this approach requires $\mathcal{O}(T \cdot 2^{|C|})$ space and time for a sequence of length $T$ and $|C|$ local properties. This exponential space and time complexity renders this approach inefficient for applications such as retrieval where sequences from a large database (with potentially many local properties) need to be checked. Secondly, this approach has no provision to handle incorrectly low or high, i.e., *potentially noisy*, scores for local properties.

We propose a novel scoring function called LogSTOP, inspired by quantitative semantics for LTL (Fainekos & Pappas, 2009), that assigns a score for a sequence of length $T$ satisfying temporal property $\varphi$ in $\mathcal{O}(T \cdot |\varphi|)$ time and space. LogSTOP employs a simple downsampling and smoothing strategy for handling locally incorrect predictions by the local property predictors. This makes it robust to cases where, for example, an object detector detects a car with very low scores in some frames because it is occluded. Moreover, the linear time computational complexity makes it an efficient solution for applications such as query matching and ranked retrieval (see example in Figure 1). For query matching, we propose a length and query-adaptive threshold which is guaranteed to accept at least as many sequences as a fixed threshold. We also demonstrate how LogSTOP can be used to rank sequences based on subsequence relevance to temporal queries in $\mathcal{O}(T^2 \cdot |\varphi|)$ time.

We evaluate LogSTOP on query matching and ranked retrieval, with sequence modalities including videos and speech, and local properties such as objects, actions, and emotions. We focus on 15 diverse temporal property templates of varying complexity. Since no existing benchmarks support this breadth of temporal properties and sequence types, we propose two new benchmarks: the QMTP (Query Matching for Temporal Properties) benchmark for objects-in-videos from the RealTLV dataset (Choi et al., 2024), and emotions-in-speech from the IEMOCAP dataset (Busso et al., 2008); and the TP2VR (Temporal Property to Video Retrieval) benchmark for objects-in-videos from the RealTLV dataset, and actions-in-videos from the AVA dataset (Gu et al., 2018).

We find that LogSTOP with simple detection models, such as YOLO and HuBERT (Hsu et al., 2021; wen Yang et al., 2021), outperforms baselines including Large Vision / Audio Language Models (LVLMs / LALMs), and NSVS-TL , on query matching by more than $16\%$ in terms of balanced accuracy. Similarly, LogSTOP with Grounding DINO (Liu et al., 2024a) and SlowR50 (Feichtenhofer et al., 2019) outperforms zero-shot text-to-video retrieval methods, such as mPLUG (Li et al., 2022) and text-text similarity with video captions, by more than $19\%$ and $16\%$ in terms of mean average precision and recall respectively.

## 2 REPRESENTING TEMPORAL PROPERTIES OVER PREDICTION SEQUENCES

Let $X = [x_1, \ldots, x_T]$ denote a sequence of data items of length $T$, where $x_t$ denotes the data item at timestep $1 \leq t \leq T$. Let $\mathcal{X}$ denote the set of all such sequences, and $\mathcal{T}$ denote the set of all timesteps $\{1, \ldots, T\}$. Further, let $C$ denote a finite set of local properties of interest. In general, $X$ and $C$ could correspond to sequences of arbitrary modalities and properties respectively, including but not limited to objects or actions in videos, and speakers or emotions in audio clips. While these sequences could be over continuous time, we assume that they are discretized into $T$ timesteps for simplicity.

We assume that there exists a true labeling function $y : \mathcal{X} \times C \times \mathcal{T} \mapsto \{0, 1\}$ such that $y(X, c, t) = 1$ if the property $c \in C$ is expressed by $x_t$, and $y(X, c, t) = 0$ otherwise. For example, $X$ could correspond to a video with frames $x_t$ at timestep $t$ and $C$ could be objects of interest such as "car" and "pedestrian". In this case, $y(X, c, t) = 1$ would indicate that object $c$ is present in frame $x_t$.

In this work, we are interested in temporal compositions of these local properties. For example, given the true labels for the object "car" in individual frames, how can we define the label for a car being present in all frames or alternatively any frame? Or, given the true labels for "car" and "pedestrian", how can we define the label for a car being present in all frames until a pedestrian is detected?

We find that Linear Temporal Logic (LTL), widely used for formal specification and verification of reactive systems (Pnueli, 1977), provides a suitable *language* for expressing such temporal properties. Formally, a temporal property $\varphi$ over local properties $C$ can be expressed in LTL as follows:

$$\varphi := \top \mid \bot \mid c \mid \neg\varphi \mid \varphi_1 \wedge \varphi_2 \mid \varphi_1 \vee \varphi_2 \mid \bigcirc\varphi \mid \Box\varphi \mid \varphi_1 \, \mathcal{U} \, \varphi_2$$

where, $c \in C$ is a local property and $\varphi_1, \varphi_2$ are temporal properties. $\neg, \wedge, \vee$ are the logical *not*, *and*, and *or* operators respectively. $\bigcirc, \Box$ and $\mathcal{U}$ are temporal operators *Next*, *Always*, and *Until* respectively. Other temporal operators such as "Eventually $\varphi$" ($\Diamond\varphi$) can then be derived as $\neg\Box\neg\varphi$.

This language can now be used to represent properties from the previous examples. For instance, the temporal properties for a car being present in all frames and any frame in a video can be represented as $\varphi = \Box\,\text{car}$ and $\varphi = \Diamond\,\text{car}$ respectively. Furthermore, the property that a car is present in all frames until a pedestrian is present can be represented as $\varphi = \text{car}\,\mathcal{U}\,\text{pedestrian}$.

The ground truth labeling function $y(X, c, t)$ for local properties can be lifted to $y(X, \varphi, t)$, meaning that the sequence $X$ expresses temporal property $\varphi$ starting at timestep $t$, using the standard semantics for LTL over finite sequences (De Giacomo & Vardi, 2013) as follows: for $1 \leq t \leq T$,

- $y(X, \top, t) = 1$ and $y(X, \bot, t) = 0$
- $y(X, \neg\varphi, t) = 1$ iff $y(X, \varphi, t) = 0$
- $y(X, \varphi_1 \wedge \varphi_2, t) = 1$ iff $y(X, \varphi_1, t) = 1$ and $y(X, \varphi_2, t) = 1$
- $y(X, \varphi_1 \vee \varphi_2, t) = 1$ iff $y(X, \varphi_1, t) = 1$ or $y(X, \varphi_2, t) = 1$
- $y(X, \bigcirc\varphi, t) = 1$ iff $t < T$ and $y(X, \varphi, t+1) = 1$
- $y(X, \Box\varphi, t) = 1$ iff $y(X, \varphi, t') = 1$ for all $t \leq t' \leq T$
- $y(X, \varphi_1 \, \mathcal{U} \, \varphi_2, t) = 1$ iff there exists a $t \leq t' \leq T$ such that $y(X, \varphi_2, t') = 1$ and $y(X, \varphi_1, t'') = 1$ for all $t \leq t'' < t'$

Given the true labeling functions for local properties $y(X, c, \cdot)$, this semantics can perfectly determine if a sequence $X$ expresses a temporal property $\varphi$ (if and only if $y(X, \varphi, 1) = 1$). In practice, however, we do not have access to these true labeling functions. We assume that noisy predictors can be used instead to provide *scores* $\hat{y} : \mathcal{X} \times C \times \mathcal{T} \mapsto [a, b]$ for the label being 1, where $a$ and $b$ are the lower and upper bounds of the score range respectively. The estimate for true label $y(X, c, t)$ can then be computed as $\tilde{y}(X, c, t) = \hat{y}(X, c, t) > \tau$ for some threshold $\tau$. Most neural models, including object detection models such as YOLO, provide scores in $[0, 1]$ and an object $c$ is said to be detected at $t$ if $\hat{y}(X, c, t) > \tau$, where $\tau$ usually is $0.5$. The accuracy of these predictors then just measures how well $\tilde{y}(X, c, \cdot)$ estimates $y(X, c, \cdot)$.

We formally introduce the problem of assigning Scores for TempOral Properties (STOPs) as follows: *Given predictors for local properties $\hat{y} : \mathcal{X} \times C \times \mathcal{T} \mapsto [a, b]$ and a temporal property $\varphi$ defined over local properties in $C$, how can a score for $\varphi$ and sequence $X$ at time step $1 \leq t \leq T$, $\hat{y}(X, \varphi, t)$, be assigned?*

---

**Algorithm 1** LogSTOP $\hat{y}(X, \varphi, t_s, t_e, w)$

---

**Input:** Sequence $X$, Temporal property $\varphi$, current timestep $1 \leq t_s \leq T$, end timestep $t_s \leq t_e \leq T$, downsampling-smoothing window $1 \leq w \leq (t_e - t_s + 1)$
**Output:** $\hat{y}(X, \varphi, t_s, t_e, w)$

 1: **function** $\hat{y}(X, \varphi, t_s, t_e, w)$
 2:   **if** $t_s > t_e$ **then return** $-\infty$
 3:   **else if** $\varphi = \top$ **then return** $0$
 4:   **else if** $\varphi = \bot$ **then return** $-\infty$
 5:   **else if** $\varphi = c$ **then**
 6:     **return** $\log(\text{avg}_{t' \in [t_s, \min\{t_s+w, t_e\}]} e^{\hat{y}(X, c, t')})$       ▷ Smooth scores in window $[t_s, t_s + w]$
 7:   **else if** $\varphi = \neg\varphi'$ **then**
 8:     **return** $\log(1 - e^{\hat{y}(X, \varphi, t_s, t_e, w)})$
 9:   **else if** $\varphi = \varphi_1 \wedge \varphi_2$ **then**       ▷ $\varphi_1$ And $\varphi_2$
10:     **return** $\hat{y}(X, \varphi_1, t_s, t_e, w) + \hat{y}(X, \varphi_2, t_s, t_e, w)$
11:   **else if** $\varphi = \varphi_1 \vee \varphi_2$ **then**       ▷ $\varphi_1$ or $\varphi_2$
12:     **return** $\hat{y}(X, \neg(\neg\varphi_1 \wedge \neg\varphi_2), t_s, t_e, w)$
13:   **else if** $\varphi = \bigcirc \varphi'$ **then**       ▷ Next $\varphi_1$
14:     **return** $\hat{y}(X, \varphi', t_s + w, t_e, w)$       ▷ Shift the current timestep from $t_s$ to $t_s + w$
15:   **else if** $\varphi = \Box \varphi'$ **then**       ▷ Always $\varphi_1$
16:     **return** $\hat{y}(X, \varphi' \wedge \bigcirc \Box \varphi', t_s, t_e, w)$
17:   **else if** $\varphi = \varphi_1 \mathcal{U} \varphi_2$ **then**       ▷ $\varphi_1$ Until $\varphi_2$
18:     **return** $\hat{y}(X, \varphi_2 \vee (\varphi_1 \wedge \neg\varphi_2 \wedge \bigcirc (\varphi_1 \mathcal{U} \varphi_2)), t_s, t_e, w)$
19:   **end if**
20: **end function**

---

## 3 LogSTOP: An Algorithm for Computing STOPs

Quantitative semantics for variants of LTL, such as Metric or Signal Temporal Logic (MTL/STL), have been proposed to quantify how well a sequence satisfies a temporal property, in $\mathcal{O}(poly(T \cdot |\varphi|))$ time. These semantics have been widely used for monitoring, falsification, and control synthesis and differ in how the degree of satisfaction is interpreted (Fainekos & Pappas, 2009; Donzé & Maler, 2010; Akazaki & Hasuo, 2015; Mehdipour et al., 2024). For instance, the standard quantitative semantics for STL, *spatial robustness*, uses the min and max operators to compute deviations from satisfaction (Fainekos & Pappas, 2009); robustness of "Always p" is the minimum score for $p$ over the sequence. We provide more details on this standard semantics in Appendix B.

Inspired by this literature on quantitative semantics, we propose a scoring function *LogSTOP*, that recursively computes a score for a sequence $X[t_s : t_e]$ satisfying temporal property $\varphi$, given start and end timesteps $t_s$ and $t_e$, and a smoothing window $w$ that we discuss later (Algorithm 1). LogSTOP provides a solution to the STOP problem with $t_s = t$ and $t_e = T$. There are three key design choices that distinguish LogSTOP from other quantitative semantics and prior work:

First, the LogSTOP for a sequence with respect to a temporal property represents the log probability of the sequence satisfying the temporal property if certain assumptions are met. Concretely, this is true when (1) the local properties represent independent events over time, (2) the scores for local properties reflect true log probabilities, and (3) temporal properties consist of compositions of independent local properties. We acknowledge that these assumptions are rarely true for real-world sequences and properties. For instance, the presence of "car" at timestep $t$ and $t + 1$ are not independent events. However, these assumptions allow us to use ideas from probability theory for independent events to compute the score. Moreover, our experiments in Section 4 show that LogSTOPs are useful for applications such as query matching and ranked retrieval even when these assumptions are not met.

Second, LogSTOP deals with potentially noisy local predictions by downsampling and smoothing predictions over windows of length $w$ (Algorithm 1, line 6). This essentially captures the property that local property scores cannot change drastically in a short local window (objects cannot momentarily disappear and reappear, actions cannot change in fractions of seconds, etc.). Note that $w$ is a hyperparameter; a higher value of $w$ can be used to control for higher variance in local predictions.

Third, LogSTOP operates in the log space to prevent underflow with fixed precision and hence assumes that the scores for local properties are given in range $[-\infty, 0]$, i.e., $\hat{y}(c, \cdot) \in [-\infty, 0]$. Whenever needed, we normalize the $\hat{y}(c, \cdot)$ to be in the $[0, 1]$ range using $e^{\hat{y}(c, \cdot)}$.

We briefly discuss how different operators are handled in Algorithm 1 and defer a detailed discussion with examples to Appendix A. The scores for logical operators, negation $\neg\varphi$, conjunction $\varphi_1 \wedge \varphi_2$, and disjunction $\varphi_1 \vee \varphi_2$ are computed using simple rules from probability theory . Concretely, LogSTOP for $\varphi_1 \wedge \varphi_2$ is the sum of the LogSTOPs for $\varphi_1$ and $\varphi_2$ (line 10), and the LogSTOP for $\varphi_1 \vee \varphi_2$ is computed using DeMorgan's law (line 12). The score for the "next" operator $\bigcirc\varphi$ is computed by shifting the timestep by one window (line 14). Scores for the "always" ($\square\varphi$) and "until" ($\varphi_1\mathcal{U}\varphi_2$) operators are computed recursively using the scores for these properties at the next window (lines 16-18). Informally, the LogSTOP for Always $\varphi$ at $t$ can be computed with a *"temporal and"* over $\varphi$ at $t$ and Always $\varphi$ at the next window, $t + w$. Similarly, the LogSTOP for $\varphi_1 \mathcal{U} \varphi_2$ can be computed with a *"temporal or"* over (1) $\varphi_2$ at $t$, and (2) $\varphi_1$ at $t$ with $\varphi_1 \mathcal{U} \varphi_2$ at the next window.

**Complexity analysis for LogSTOP.** The computational complexity of Algorithm 1, for a temporal property $\varphi$ with length $|\varphi|$ and a sequence of $T$ predictions is $\mathcal{O}(T \cdot |\varphi|)$. This uses dynamic programming to cache scores for all sub-properties over the sequence (Fainekos et al., 2012). The key observation here is that at any timestep $t$, the LogSTOP for *any* property $\varphi$ can be computed in $\mathcal{O}(|\varphi|)$ given the LogSTOPs for its sub-properties at $t$ and itself at $t + w$. This is because the LogSTOPs for temporal properties are defined recursively and there are at most $|\varphi|$ sub-properties. For $T/w$ timesteps, this results in $\mathcal{O}((T/w) \cdot |\varphi|)$ time. Since the smoothing operation takes $\mathcal{O}(w)$ time per window, computing LogSTOP for $\varphi$ over a sequence of length $T$ requires $\mathcal{O}(T \cdot |\varphi|)$ time.

## 3.1 LogSTOP for Query Matching

We define query matching with temporal properties as the task of predicting whether a given temporal property / query matches, or is expressed by, a sequence. LogSTOPs can be used for matching sequence $X$ with query $\varphi$ by comparing $\hat{y}(X, \varphi, 1, T)$ with an appropriate threshold.

A natural first choice for such a threshold for LogSTOP is the constant $\tau = \log 0.5$. This threshold, is employed by existing works to determine if a video satisfies a temporal property (Yang et al., 2023; Choi et al., 2024). This, however, does not scale with the length of the sequence. For instance, given a 6-frame video with constant $\log 0.9$ scores for "car", the LogSTOP for temporal property "Always car", with $w = 1$, is $\log(0.9^6)$. This is greater than $\log(0.5)$ and hence the video matches the query. However, when another frame with the same high score $\log(0.9)$ is added, the score drops to $\log(0.9^7)$, which is less than $\log(0.5)$ and hence the video no longer matches the query. We would ideally also like the latter to match the query since the property "car" is detected with high scores.

We propose an adaptive threshold $\tau$ for query $\varphi$ and sequence length $T$ as follows:

$$\hat{y}_{0.5}(\cdot, \varphi, T, w) = \hat{y}(\cdot, \varphi, 1, T, w) \text{ using } \hat{y}_{0.5}(\cdot, c, t) = \log 0.5 \text{ for all } c \in C, 1 \le t \le T$$
$$\tau(\varphi, T, w) = \min\{\log 0.5, \ \hat{y}_{0.5}(\cdot, \varphi, T, w)\}$$

Informally, a sequence expresses a temporal property if the LogSTOP is higher than both random chance $\log 0.5$ and LogSTOP using random chance predictors for local properties $\hat{y}_{0.5}(\cdot, \varphi, T, w)$. This threshold can be computed in $\mathcal{O}(T \cdot |\varphi|)$ and is guaranteed to match at least as many sequences as the constant $\log 0.5$ threshold. For properties where the score decreases with sequence length (e.g., $\square\varphi$), the adaptive threshold allows more sequences to match the query than the constant threshold.

## 3.2 LogSTOP for Ranked Retrieval

LogSTOP can also be used for the task of ranking and retrieving sequences relevant to temporal properties of interest. Formally, given a database of $N$ sequences $\mathcal{D} = \{X_1, \ldots, X_N\}$ a temporal property $\varphi$, and a range of event lengths $(T_{lo}, T_{hi})$, the goal is to rank each $X_i$ based on whether it contains a subsequence $X_i[t : t']$ of length $t' - t \in [T_{lo}, T_{hi}]$ that expresses $\varphi$. Examples of such queries include "videos with a 10 to 20 second scene where a person is sitting down until they stand up". This task is different from the query matching task in two key ways: firstly, the relative ranking of sequences is more important than absolute scores. Secondly and more importantly, the relevance of a sequence may only be with respect to a part of the sequence (a *moment* in the video, for example).

Algorithm 2 outlines how LogSTOP can be used for ranked retrieval. Informally, given a temporal property $\varphi$, sequence $X$, and event duration $(T_{lo}, T_{hi})$, the relevance of $X$ to $\varphi$ is defined as the maximum LogSTOP of any subsequence of $X$ of length in $[T_{lo}, T_{hi}]$. The relevance score for any sequence can be computed in $\mathcal{O}(T^2 \cdot |\varphi|)$ time since LogSTOPs for suffix subsequences are cached

---

**Algorithm 2** LogSTOP for Ranked Sequence Retrieval

---

**Input:** Database $\mathcal{D} = \{X_1, X_2, \ldots, X_N\}$, temporal property $\varphi$, event length range $(T_{lo}, T_{hi})$, number of retrievals $k$, smoothing window for LogSTOP $w$
**Output:** Ranked list $\mathcal{R}$
1: $R \leftarrow []$
2: **for** $X_i \in \mathcal{D}$ **do**
3:    $s_i \leftarrow -\inf$
4:    $T \leftarrow X_i.length()$
5:    **for** $T_{end} \in \{T_{lo}, \ldots, T\}$ **do**
6:       $T_{start} \leftarrow \max\{1, T_{end} - T_{hi}\}$            ▷ Max length of subsequence is $T_{hi}$
7:       Compute $\hat{y}(X_i, \varphi, T_{start}, T_{end}, w)$ using Algorithm 1   ▷ Caches scores for suffix subsequences
8:       $s_{i,T_{end}} \leftarrow \max\{\hat{y}(X_i, \varphi, t, T_{end}, w)$ for $t \in [T_{start}, T_{end} - T_{lo}]\}$
9:       $s_i \leftarrow \max(s_i, s_{i,T_{end}})$                    ▷ Track maximum score
10:    **end for**
11:    $\mathcal{R}.append((i, s_i))$
12: **end for**
13: **return** top-$k$ sequences from $\mathcal{R}$ in decreasing order of $s_i$

---

with dynamic programming. Note that this represents one way of computing scores for ranking videos, where subsequences of certain lengths are relevant to queries; there could be other variants which LogSTOP could be used for but are not considered (for example, computing the score with respect to the entire video, or only considering videos where the subsequence score is over a threshold).

## 4 EXPERIMENTS

We select 15 temporal property templates from 5 broad categories for evaluation, in the order of increasing difficulty of operator selection and nesting (p1, p2, p3 are placeholders for local properties):

1. **Simple temporal operators:** Eventually p1, Always p1, p1 Until p2.

2. **Boolean over temporal operators:** Always p1 and Eventually p2, Always p1 or Eventually p2.

3. **Temporal over boolean operators:** (Not p1) Until p2, p1 Until (Not p2), Always (p1 and p2), (p1 and p2) Until p3.

4. **Temporal over temporal operators:** p1 Until Always p2, Eventually Always p1, Always Eventually p1.

5. **Temporal operators over boolean and temporal operators:** (Not p1) Until Eventually p2, (Not p1) Until Always p2, (p1 and p2) Until Eventually p3.

### 4.1 THE QMTP AND TP2VR BENCHMARKS

There are no existing benchmarks that evaluate query matching and ranked retrieval on video and speech sequences with the breadth of temporal properties discussed above. We hence introduce two new benchmarks for evaluation using three existing datasets with frame/segment-level annotations for local properties: RealTLV (Choi et al., 2024) for objects in videos (6 classes), IEMOCAP (Busso et al., 2008) for emotions in speech (4 classes), and AVA (Gu et al., 2018) for actions in videos (80 classes). We briefly describe the two benchmarks below, with more details in Appendix C.

**The QMTP benchmark.** The QMTP benchmark evaluates query matching with temporal properties over objects in video and emotions in speech sequences. **QMTP-video** consists of 7468 samples (3750 matching and 3718 non-matching) with $10 - 50$ frames per sample. **QMTP-speech** contains 3300 samples (balanced), including speech sequences with $5 - 30$ segments per sample.

**The TP2VR benchmark.** The TP2VR benchmark evaluates ranked retrieval of video sequences given temporal property queries over objects and actions. The **TP2VR-objects dataset** consists of 746 videos with $39 - 199$ frames, collected from the RealTLV dataset, and 42 queries over objects. Each query corresponds to $25 - 50$ frame temporal events and is relevant to no more than 250 videos in the dataset. Similarly, the **TP2VR-actions dataset** consists of 952 videos with 300 frames each, collected from 1-min segments of videos in the AVA dataset, with 70 queries over actions. Each query corresponds to 10-second temporal events and is relevant to no more than 50 videos in the dataset.

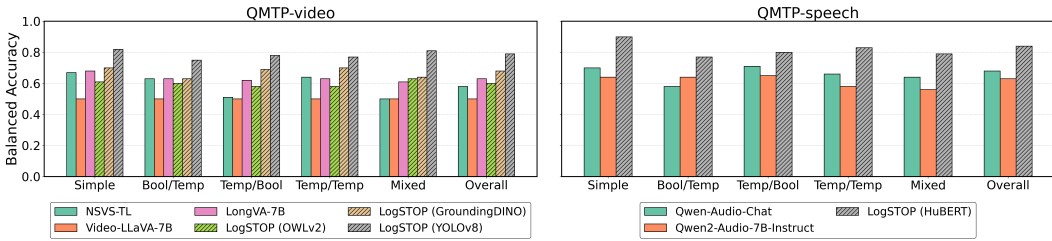

Figure 2: LogSTOP outperforms other methods on the QMTP-video and QMTP-speech datasets. The average balanced accuracy for the five temporal property categories and overall is presented. Detailed results for all queries are provided in Appendix J.2.

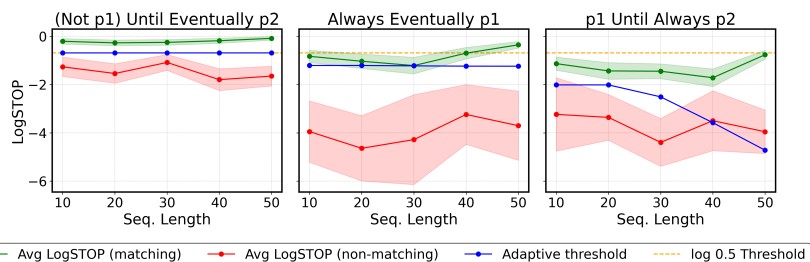

Figure 3: The adaptive threshold accepts more matching sequences than the constant $\log 0.5$ threshold. LogSTOPs with YOLOv8 (mean with 95% CI) are shown for sequences from QMTP-video. Comparison is shown for three properties, with results for other properties in Appendix I.1.

## 4.2 RESULTS ON QUERY MATCHING

**Methods.** We evaluate LogSTOP for temporal query matching using simple neural predictors for object and emotion detection. We use YOLOv8 (Jocher et al., 2023), OWLv2 (Minderer et al., 2023), and Grounding DINO (Liu et al., 2024a) to obtain frame-level object detection scores. We use HuBERT (wen Yang et al., 2021) for segment-level emotion recognition. These scores are matched using the adaptive threshold discussed in Section 3.1. For QMTP-video, we compare against two Large Vision Language Models (LVLMs), namely `Video-LLava-7B` (Lin et al., 2023) and `LongVA-7B` (Zhang et al., 2024a), and the PCTL-based method, NSVS-TL (Choi et al., 2024). For QMTP-speech, we compare against two Large Audio Language Models (LALMs), namely `Qwen-Audio-Chat` (Chu et al., 2023) and `Qwen2-Audio-7B-Instruct` (Chu et al., 2024). We provide more details on the prompts and parameters used for all methods in Appendix D.

**Results.** Figure 2 shows the balanced accuracies of different methods on the QMTP-video and QMTP-speech datasets. LogSTOP outperforms other methods by at least $16\%$ on QMTP-video and QMTP-speech using object detection scores from YOLOv8 and emotion detection scores from HuBERT respectively. LogSTOP with Grounding DINO also performs better than the baselines. The accuracies of detecting objects with scores $> 0.5$ for YOLO, Grounding DINO and OWLv2 are $46\%$, $38\%$ and $19\%$ resp. which reflect the order of their performances on query matching.

LogSTOP consistenty reports accuracies over $75\%$ on all query categories. `LongVA-7B` and `Qwen-Audio-Chat` perform better on simple temporal queries than queries with boolean / temporal compositions. NSVS-TL also performs poorly on categories with compositions over boolean expressions. These results suggest that understanding of temporal queries is still an open problem for LVLMs and LALMs. Moreover, the higher accuracy of LogSTOP with much smaller neural models suggests that using well-defined logics for reasoning is beneficial.

Finally, we evaluate how the various design choices for LogSTOP affect the performance (Table 1). We find that the accuracy drops by $2\%$ when the standard STL robustness is used for aggregating scores instead of LogSTOP, or when local smoothing from Algorithm 1 (line 6) is not performed. A $3\%$ drop is also observed when the adaptive threshold is replaced with $\log 0.5$; Figure 3 demonstrates how the adaptive threshold is better at distinguishing between matching and non-matching sequences.

Table 1: Performance of LogSTOP on query matching and ranked retrieval drops as components (aggregation method, smoothing, threshold) are ablated. Results are shown on QMTP-video (with YOLOv8) and TP2VR-objects (with GroundingDINO). STL robustness is described in Appendix B.

| Ablation | QMTP-video | TP2VR-objects | | |
|---|---|---|---|---|
| | *Balanced Accuracy* | *mAP* | *R@r* | *MnR* |
| *LogSTOP* | 0.79 ($\downarrow$ 0%) | 0.64 ($\downarrow$ 0%) | 0.59 ($\downarrow$ 0%) | 2.0 ($\downarrow$ 0) |
| *Replace LogSTOP with STL Robustness* | 0.77 ($\downarrow$ 2%) | 0.52 ($\downarrow$ 12%) | 0.45 ($\downarrow$ 14%) | 3.8 ($\uparrow$ 1.8) |
| *LogSTOP without local smoothing* | 0.77 ($\downarrow$ 2%) | 0.59 ($\downarrow$ 5%) | 0.55 ($\downarrow$ 4%) | 1.8 ($\downarrow$ 0.2) |
| *LogSTOP with* ($\log 0.5$) *threshold* | 0.76 ($\downarrow$ 3%) | - | - | - |

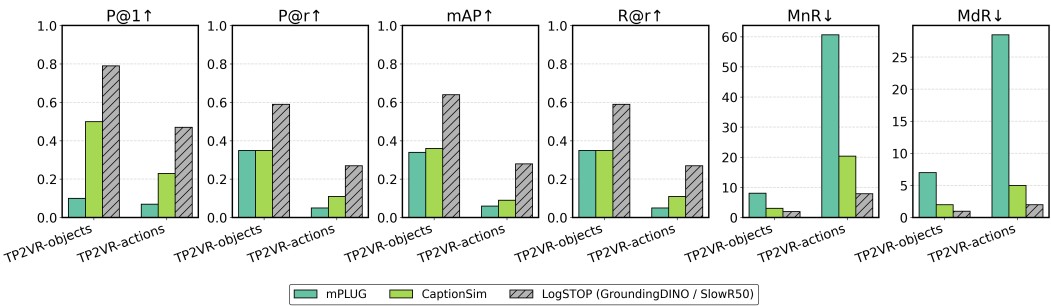

Figure 4: LogSTOP outperforms zero-shot text-to-video retrieval methods on the TP2VR benchmark ($r$ denotes the number of relevant sequences). Detailed results are in Appendix J.1.

## 4.3 RESULTS ON RANKED RETRIEVAL

**Methods.** We evaluate LogSTOP for ranked retrieval using Grounding DINO for object detection and Detectron2 (Wu et al., 2019) with SlowR50 (Feichtenhofer et al., 2019) for action detection. Since there are no methods specifically designed for temporal property to sequence retrieval, we adapt existing text-to-video retrieval methods for this task. Since LogSTOP does not require explicit training for retrieval, we specifically only include zero-shot text-to-video retrieval methods for comparison. We include mPLUG (Li et al., 2022), a large multimodal model that jointly embeds videos and text queries. Inspired by ELIOT (Liu et al., 2025), we also include embedding similarity between video captions and text queries. We refer to this as CaptionSim and use LLaVA-NeXT-Video-7B (Zhang et al., 2024b) for generating video captions and SentenceBERT (Reimers & Gurevych, 2020) for embedding captions and queries. More details are provided in Appendix E.

**Metrics.** Following existing work, we include standard retrieval metrics such as Recall at $r$ (R@$r$, where $r$ is the number of relevant results) for evaluating coverage, and mean / median ranks of first retrieval (MnR / MdR). Since multiple videos could be relevant to a query, we also evaluate if relevant results are ranked higher using Precision (P@$\{1, r\}$), and mean average precision (mAP).

**Results.** Figure 4 presents the results for ranked retrieval on the TP2VR benchmark. The performance of all methods on TP2VR-actions is lower than that on TP2VR-objects due to the significantly higher number of classes (80 actions vs. 6 objects) and lower number of relevant results (on average, 21 vs. 163 relevant videos). LogSTOP with GroundingDINO outperforms mPLUG and CaptionSim on TP2VR-objects by at least 28% in mAP and 24% in R@$r$, indicating that relevant results are retrieved at earlier ranks and the retrieved results include more relevant items than other methods. Similarly, LogSTOP with SlowR50 outperforms baselines by more than 19% and 16% in terms of mAP and R@r respectively. The first relevant result is also retrieved earlier by LogSTOP as is indicated by at least a 24% higher P@1 and better mean ranks; on TP2VR-actions, LogSTOP retrieves the first relevant result at rank 7.9 while CaptionSim and mPLUG retrieve it at ranks > 20. Figure 5 presents examples of ranking using the three methods. We discuss these in detail in Appendix H.

Ablating various components of LogSTOP also degrades retrieval performance (Table 1). The standard STL robustness reduces mAP and R@$r$ by more than 12%. While removing the smoothing step from LogSTOP leads to a slight increase of 0.2 in MnR, it reduces mAP and R@$r$ by at least 4%.

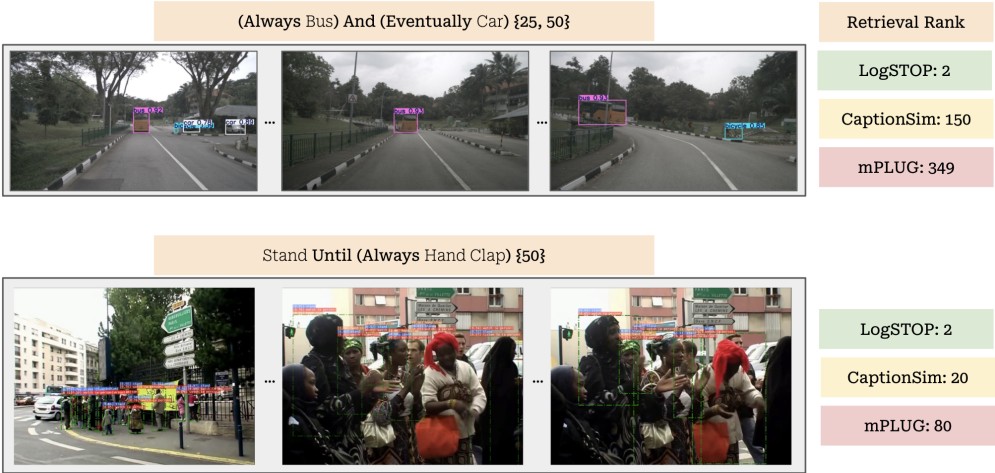

Figure 5: Examples of video retrieval with different methods, from the TP2VR-objects and TP2VR-actions datasets. The event length ranges in terms of number of frames are mentioned with the temporal properties. Detailed discussion of these examples is in Appendix H

## 5    RELATED WORK

**Temporal Logic for video and audio understanding.**    Yang et al. (2023) and Choi et al. (2024) (NSVS-TL) use the probabilistic model checker STORM to verify temporal properties over object detections in videos, using LTL and PCTL to represent properties respectively. As discussed in the introduction, the key limitation of these approaches is the exponential complexity of scoring or matching, rendering them impractical for high-volume tasks such as retrieval. Morever, these methods do not explicitly handle noisy local predictions, and use the constant $0.5$ threshold for matching. We propose LogSTOP as an efficient linear-time alternative to these methods with local smoothing for handling noisy predictions and an adaptive threshold for matching.

**Benchmarks for video and audio understanding.**    Benchmarks for video understanding such as Video-MME (Fu et al., 2024), RexTIME (Chen et al., 2024), Next-qa (Xiao et al., 2021), QVHighlights (Lei et al., 2021), TemporalBench (Cai et al., 2024) and TempCompass (Liu et al., 2024b) include tasks that require temporal understanding of events in videos. Similarly, audio understanding datasets such as MMAU (Sakshi et al., 2024) and CompA (Ghosh et al., 2023) evaluate temporal tasks such as detecting the order of two events. These tasks are fundamentally different from the QMTP benchmark which focuses on more fine-grained temporal properties.

**Video retrieval with temporal queries.**    Popular text-to-video retrieval datasets such as Activity Net Captions (Krishna et al., 2017) and DiDeMo (Anne Hendricks et al., 2017) focus on temporal segments within minute-long videos. Our TP2VR benchmark focuses on fine-grained temporal queries over short events in videos, with many-to-many mapping between queries and videos.

Popular text-video retrieval methods include CLIP4Clip (Luo et al., 2021), TS2-Net (Liu et al., 2022), (Bain et al., 2021), which employ training to improve embeddings for retrieval, and zero-shot methods such as mPLUG (Li et al., 2022) and ELIOT (Liu et al., 2025). Since we use off-the-shelf models with LogSTOP for retrieval, we only include the latter for comparison.

## 6    CONCLUSION, LIMITATIONS AND FUTURE WORK

In this work, we present the problem of assigning scores for temporal properties (STOPs) given potentially noisy score predictors for local properties. We represent these properties using LTL and propose a scoring function LogSTOP for assigning STOPs. We then introduce the QMTP and TP2VR benchmarks for evaluating query matching and ranked retrieval with temporal properties over objects / actions in videos and emotions in speech. LogSTOP with simple neural predictors outperforms LVLMs / LALMs, Temporal Logic-based baselines, and text-to-retrieval methods on the benchmarks.

**Limitations.** There are properties such as "there are always 2 cars" that cannot directly be expressed in LTL. Future work should hence explore more expressive logics (Anderson et al., 2023; Huang et al., 2023) or construct local predictors for complex properties. While we only focus on sequences with single modalities, it will be interesting to see LogSTOP being for multi-modal applications where the local properties are over different modalities with scores from different local predictors.

**Reproducibility statement.** LogSTOP is described clearly in Algorithm 1, along with a detailed discussion on how it can be used for query matching and ranked retrieval (Algorithm 2). The QMTP and TP2VR benchmarks use videos and speech clips (along with annotations) from existing publicly available datasets. We provide details of benchmark construction and processing in Appendix C. As base local property predictors for LogSTOP, we use off-the-shelf publicly available object detection, emotion recognition, and action detection models. As baselines, we use openly available LVLMs / LALMs and official implementations of NSVS-TL and mPLUG. Appendix D and Appendix E desribe all the methods used in detail. We list queries and prompts for all methods in Appendix F. We will open-source the benchmarks and code for using LogSTOP for query matching and retrieval upon acceptance.

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

## A    MORE DETAILS ON ALGORITHM 1

Algorithm 1 describes how LogSTOP is computed for a sequence $X[t_s : t_e]$ with respect to temporal property $\varphi$, given start and end timesteps $t_s$ and $t_e$, and smoothing window $w$. Here we discuss the intuition behind the operators, along with some examples:

**Downsampling and average smoothing of confidence scores of local properties.** Firstly, LogSTOP downsamples the sequence of length $T$ to contiguous blocks of length $w$. The confidence scores for any local property in each block starting at $t$, $\hat{y}(c, t' \in [t, t + w])$ are first averaged after being normalized to the $[0, 1]$ range. The confidence score for each block is then the $\log$ of this averaged $[0, 1]$-normalized score. This series of downsampling, normalizing and smoothing operations starting at timestep $t$ for a window $w$ can be seen on line 6. For example, given a sequence of log scores for object "car",

$$\hat{y}(car, t \in [0, 5]) = [\log(0.9), \log(0.1), \log(0.9), \log(0.9), \log(0.9), \log(0.9)]$$

and $w = 3$, the LogSTOP first downsamples to 2 blocks,

$$[[\log(0.9), \log(0.1), \log(0.9)], [\log(0.9), \log(0.9), \log(0.9)]]$$

before averaging by block,

$$[\log((0.9 + 0.1 + 0.9)/3), \log((0.9 + 0.9 + 0.9)/3)] = [\log(0.63), \log(0.9)]$$

In this example, note that the score $\log(0.1)$ is likely incorrect since the car cannot momentarily disappear. Downsampling and then smoothing reduce the impact of this incorrect local prediction hence essentially capturing the property that confidence scores cannot drastically change in a local window. Note that this is done online for each successive block and the shifting for temporal operators is handled by the Next operator (line 14).

**Handling of logical operators ($\neg, \wedge, \vee$).** The LogSTOP for $\neg\varphi$ at timestep $t$ is intuitively high when the score for $\varphi$ is low (line 8). For example, given a high score $\hat{y}(car, t, w) = \log(0.9)$, the score for "not car" $\hat{y}(\neg car, t, w) = \log(1 - 0.9) = \log(0.1)$ is low.

The LogSTOP for $\varphi_1 \wedge \varphi_2$ at timestep $t$ is high only when the scores for both $\varphi_1$ and $\varphi_2$ are high (line 10). For example, given high scores $\hat{y}(car, t, w) = \log(0.9)$ and $\hat{y}(pedestrian, t, w) = 0.9$, the score for "car and pedestrian" $\hat{y}(car \wedge pedestrian, t, w) = \log(0.9) + \log(0.9) = \log(0.81)$. However, if either of the scores are low, e.g., if $\hat{y}(pedestrian, t, w) = 0.1$, the score drops significantly to $\hat{y}(car \wedge pedestrian, t, w) = \log(0.9) + \log(0.1) = \log(0.09)$. Inspired by DeMorgan's law, the LogSTOP for $\varphi_1 \vee \varphi_2$ is simply the score for equivalent property $\neg(\neg\varphi_1 \wedge \neg\varphi_2)$ (line 12). This is intuitively only low when both the scores are low.

Note that these operators are defined not only over local properties $c$ as in the examples above, but over any temporal property $\varphi, \varphi_1, \varphi_2$. Hence, for temporal property $\varphi$ = "(car or truck) and (not pedestrian)", the score is high only if it is high for both $\varphi_1$ = "(car or truck)" and $\varphi_2$ = "(not pedestrian)". The scores for $\varphi_1, \varphi_2$ can be recursively computed.

**Handling of temporal operators ($\bigcirc, \square, \mathcal{U}$).**    As discussed above, the property Next $\varphi$ ($\bigcirc\varphi$) evaluates whether $\varphi$ is expressed starting at the next block $t + w$ (line 14). When $w = 1$, this represents the standard Next operator. The property Always $\varphi$ ($\square\varphi$) is interpreted here as a "temporal and" operator over the sequence (line 16). Hence, $\square\varphi$ can be equivalently written as $\varphi \wedge \bigcirc\square\varphi$: the property $\varphi$ is expressed by $x_t$ and always after by $[x_{t+w}, \ldots]$ ($\bigcirc\square\varphi$). Similar to the logical $\wedge$ operator, the score is high only if it is high for all timesteps of the sequence. The computation in $\log$ space is beneficial to prevent any underflow here with fixed precision.

The property $\varphi_1$ Until $\varphi_2$ ($\varphi = \varphi_1\,\mathcal{U}\,\varphi_2$) can be equivalently written as $\varphi = \varphi_2 \vee (\neg\varphi_2 \wedge \varphi_1 \wedge \bigcirc(\varphi_1\,\mathcal{U}\,\varphi_2))$ (line 18). This informally translates to evaluating whether either (1) $\varphi_2$ is expressed by $x_t$, or (2) $\varphi_1$ is expressed instead and $\varphi = \varphi_1\,\mathcal{U}\,\varphi_2$ is expressed by $[x_{t+w}, \ldots]$.

## B    MORE DETAILS ON QUANTITATIVE SEMANTICS FOR TEMPORAL LOGIC

We now present more details on the standard quantitative semantics for Signal Temporal Logic (STL) discussed in Section 3. A formula in Signal Temporal Logic can be written as:

$$\varphi := \top \mid \mu \mid \neg\varphi \mid \varphi_1 \wedge \varphi_2 \mid \varphi_1 \, \mathcal{U}_I \, \varphi_2$$

where $\mu = f(s(t)) \geq 0$ is a Lipschitz continuous function over the signal $s$ and $I = [t_1, t_2]$ is a time interval, $t_2 \geq t_1 \geq 0$. The operators *Eventually* and *Always* can be defined as follows:

$$\diamondsuit_I \varphi := \top \, \mathcal{U}_I \, \varphi$$
$$\square_I \varphi := \neg\diamondsuit_I \neg\varphi$$

In the context of the STOP problem, $s(t) = \hat{y}(X, \cdot, t)$ and $f_c(s(t)) = \hat{y}(X, c, t) - \tau$. The query "car until pedestrian" can be written as $\mu_{car}(s(t)) \mathcal{U}_I \mu_{pedestrian}(s(t))$ where $I = [0, T]$ and $\mu_{car}(s(t)) = \hat{y}(X, car, t) - \tau \geq 0$ ($\mu_{pedestrian}$ is defined similarly).

The STL quantitative semantics, also called *robustness* $\rho$ (Fainekos & Pappas, 2009), is defined as follows to indicate how much a signal satisfies or violates the formula:

$$\rho(\top, s, t) := \rho_\top$$
$$\rho(\mu, s, t) := f(s(t))$$
$$\rho(\neg\varphi, s, t) := -\rho(\varphi, s, t)$$
$$\rho(\varphi_1 \wedge \varphi_2, s, t) := \min(\rho(\varphi_1, s, t), \rho(\varphi_2, s, t))$$
$$\rho(\varphi_1 \vee \varphi_2, s, t) := \max(\rho(\varphi_1, s, t), \rho(\varphi_2, s, t))$$
$$\rho(\varphi_1 \, \mathcal{U}_I \, \varphi_2, s, t) := \sup_{t' \in t+I} (\min\{\rho(\varphi_2, s, t'), \inf_{t'' \in [t,t']} \rho(\varphi_1, s, t'')\})$$
$$\rho(\square_I \varphi, s, t) := \inf_{t' \in [t+I]} \rho(\varphi, s, t')$$
$$\rho(\diamondsuit_I \varphi, s, t) := \sup_{t' \in [t+I]} \rho(\varphi, s, t')$$

where, $\rho_\top$ is the maximum robustness, i.e., $\rho_\top = b - \tau$ for the CSTOP problem where $b = \max(\hat{y}(\cdot, \cdot, \cdot))$.

We argue that LogSTOP offers advantages over such semantics in the context of the STOP problem. This is primarily because the traditional robustness measure is defined using $\max$ and $\min$ functions over temporal and logical formulae. The measure, hence, only reflects the most violating or most satisfying timestep in the sequence. For example, consider assigning confidence scores to the property "Always car" in two different scenarios:

$$\hat{y}_1(car, t \in [0, 2]) = [\log(0.9), \log(0.9), \log(0.1)]$$
$$\hat{y}_2(car, t \in [0, 2]) = [\log(0.1), \log(0.1), \log(0.1)]$$

Ideally, the confidence score for "Always car" should follow the order: $\hat{y}_1(\square car, \cdot) > \hat{y}_2(\square car, \cdot)$. The standard STL semantics, however, would assign the same robustness to both sequences for any $\tau > 0.1$ since the most violating score is $\log(0.1)$ in either case. This makes the robustness measure unsuitable for downstream applications that require such ordering: for example, ranking / search.

**An example with Boolean operators.** For example, consider assigning confidence scores to the property "car and pedestrian" at $t = 0$ in two different scenarios:

$$\hat{y}_1(car, t = 0) = \log(0.9), \hat{y}_1(pedestrian, t = 0) = \log(0.6)$$
$$\hat{y}_2(car, t = 0) = \log(0.6), \hat{y}_2(pedestrian, t = 0) = \log(0.6)$$

Ideally, the confidence scores for "car and pedestrian" should follow the order: $\hat{y}_1(car \wedge pedestrian, t = 0) > \hat{y}_2(car \wedge pedestrian, t = 0)$ since $\hat{y}_1(car, t = 0) > \hat{y}_2(car, t = 0)$. The robustness for both the cases is the same, i.e., $\log(0.6) - \log(0.5)$, because of the $\min$ semantics for the Boolean *and* operator. The LogSTOP for the two cases are $\log(0.9) + \log(0.6)$ and $\log(0.6) + \log(0.6)$ respectively, which reflect the expected order.

**An example with the Until operator.** Consider assigning confidence scores to the property "car Until pedestrian" in two different scenarios:

$$\hat{y}_1(car, t \in [0, 2]) = [\log(0.6), \log(0.6), \log(0.6)]$$

$$\hat{y}_1(pedestrian, t \in [0, 2]) = [\log(0.4), \log(0.4), \log(0.9)]$$

and,

$$\hat{y}_2(car, t \in [0, 2]) = [\log(0.6), \log(0.6), \log(0.6)]$$

$$\hat{y}_2(pedestrian, t \in [0, 2]) = [\log(0.4), \log(0.4), \log(0.6)]$$

Note that the only difference between the two scenarios is the score for "pedestrian" at $t = 2$ (a high score of $0.9$ for the first scenario and a lower score of $0.6$ for the second scenario) . The robustness for the two cases is the same because of the $\min$ semantics within the *Until* operator. LogSTOP assigns a higher score for the first scenario because of the difference at $t = 2$.

## C   MORE DETAILS ON DATASETS

In Section 4, we briefly discuss the QMTP and TP2VR benchmarks for evaluation. For constructing these benchmarks , we use three existing datasets with frame/segment-level annotations for local properties: The RealTLV dataset (Choi et al., 2024) consists of videos from NuScenes (Caesar et al., 2020) and Waymo (Sun et al., 2020) driving datasets with frame-level annotations for 6 object classes. (for example, "car", "truck", etc). The IEMOCAP dataset (Busso et al., 2008) provides speech segments from conversations between two speakers and each segment is labeled with one of the 4 major emotions expressed by the speaker. (for example, "happy", "sad", etc.). The AVA dataset (Gu et al., 2018) consists of frame-level action annotations for 80 actions in 15-min clips from YouTube. We only consider the validation subset of this dataset and sample 5 frames per second. These datasets can be used for evaluating temporal properties over objects in videos ("car until pedestrian", for example) , emotions in speech ("always happy") , and actions in videos ("a person sits until they stand up") respectively.

**The QMTP dataset.** For any temporal property template (for example, "p1 Until p2") and samples from these datasets, we identify matching and non-matching sequences of desired length as follows: for every sample, we first identify candidates for local properties in the template (p1, p2, etc.) as the set of all ground-truth objects / emotions in the sequence. We then use the standard LTL semantics over the frame/segment-level ground-truth labels to collect matching and non-matching subsequences of the desired length. This creates a TP-query matching dataset for an arbitrary set of temporal properties as long as these properties are sufficiently expressed by sequences from the underlying dataset. Moreover, this pipeline is agnostic to the choice of the dataset since it only requires sequences of ground-truth labels for local properties. We use this pipeline to create the **QMTP-video dataset** with 7468 samples (3750 matching and 3718 non-matching) with video sequences of lengths $\{10, 20, 30, 40, 50\}$. For each target length, this dataset contains approximately 100 samples corresponding to each of the 15 property templates. Similarly, we create the **QMTP-speech dataset** with 3300 samples, including speech sequences of lengths in ranges $\{5 - 10, 10 - 20, 20 - 30\}$. The QMTP-speech dataset only contains samples from $11/15$ property templates. This is because there are no sequences matching 4 properties "Always p1 and Eventually p2", "Always (p1 and p2)", "(p1 and p2) Until p3", and "(p1 and p2) Until Eventually p3" since two emotions cannot be expressed at the same time.

**The TP2VR dataset.** We restrict queries to a maximum of 5 per temporal property template. For TP2VR objects, we aim to find $25 - 50$ frame-long subsequences satisfying a temporal property; we only include a temporal property if less than 250 videos are retrieved using the standard LTL semantics with the ground truth labels. With all possible combinations of objects, this gives us a total of 42 queries, with an average of 163 videos relevant to a query. Similarly, for TP2VR-actions, we aim to find 10-second long subsequences (50 frames) satisfying a temporal property; we only include a temporal property if less than 50 videos are retrieved using the standard LTL semantics with the ground truth labels. With all possible combinations of actions, this gives us a total of 70 queries, with an average of 21 videos relevant to a query.

## D   MORE DETAILS ON QUERY MATCHING METHODS

**LogSTOP with simple trained neural predictors:** We use YOLOv8 (Jocher et al., 2023), Grounding DINO (Liu et al., 2024a), and OWLv2 (Minderer et al., 2023), to get confidence scores for object detection in video frames, and HuBERT (wen Yang et al., 2021) for speech emotion recognition

in audio segments. Since the scores are in the $[0, 1]$ range, we normalize them in the $[-\infty, 0]$ range using the $\log$ operation, as required by LogSTOP. A video matches query $\varphi$ if LogSTOP $\hat{y}(\varphi, 0, w) > \tau(\varphi, 0)$ (and vice versa for non-matching examples). The estimates are evaluated against ground-truth labels $y(\varphi, 0)$. The window $w$ is selected as follows: $w = 2$ for $T < 20$ and $w = 5$ otherwise.

**Large Vision Language Models (LVLMs)**. We evaluate two popular LVLMs on query matching for videos: `Video-LLava-7B` (Lin et al., 2023) and `LongVA-7B` (Zhang et al., 2024a). For the "always car" example, we provide the models with the video sequence and a text prompt "Is a car detected in all frames of this video?". The response is considered correct if the model responds with "Yes" or "No" for matching and non-matching samples respectively. `Video-LLava-7B` supports a context window of $4096$ tokens while `LongVA-7B` can handle up to 2000 frames. We set the maximum tokens to generate to $60$ and $1024$ respectively and use a temperature of 0.1 and standard values for the other parameters.

**Large Audio Language Models (LALMs)**. Similarly, we evaluate two popular LALMs on query matching for speech: `Qwen-Audio-Chat` (Chu et al., 2023) and `Qwen2-Audio-7B-Instruct` (Chu et al., 2024). For the "eventually happy" example, we provide the models with the audio sequence and a text prompt "Does the speaker sound happy at some time in this audio clip?". We set the sampling rate to 16000 and generate a maximum of 256 new tokens, with standard values for other parameters.

**NSVS-TL (Choi et al., 2024).** Proposed for event detection in videos, NSVS-TL (Choi et al., 2024) uses the PCTL-based model checker STORM (Hensel et al., 2022) to identify video frame subsequences where a certain event is detected. NSVS-TL reports state-of-the-art performance on detecting temporal events in videos, surpassing large language models such as GPT-4. For our task, we specify the target query in PCTL ("always car" is $P > 0.5[G \text{ "car" }]$) and the response is considered correct if NSVS-TL returns / does not return the entire video sequence as output for a matching / non-matching query respectively.

We do not evaluate the method from Yang et al. (2023) since the implementation is not publicly available and LTL model checking with STORM is not well-documented.

# E    More details on the Ranked Retrieval methods

**LogSTOP.** We use LogSTOP with Grounding DINO (Liu et al., 2024a) for TP2VR-objects and with SlowR50 (Feichtenhofer et al., 2019) for TP2VR-actions respectively. We repurpose the script from `tutorials/video_detection_example` at `https://github.com/facebookresearch/pytorchvideo/` to run SlowR50 on videos from the TP2VR-actions dataset. We use a smoothing window $w = 5$ for all retrieval experiments.

**mPLUG.** We use the implementation from `https://github.com/alibaba/AliceMind`. We repurpose the `mPLUG/retrieval_vid_mplug.py` script to run `mPLUG_large_v2` on videos and queries from the TP2VR datasets.

**CaptionSim.** Inspired by ELIOT (Liu et al., 2025), we also include embedding similarity between video captions and text queries as a baseline. We refer to this as `CaptionSim` in the discussion, and use `LLaVA-NeXT-Video-7B` (Zhang et al., 2024b) for generating video captions and `SentenceBERT/all-MiniLM-L6-v2` (Reimers & Gurevych, 2020) for embedding captions and queries. Due to the limited context window of `LLaVA-NeXT-Video-7B`, we divide videos in sections of 50 frames and generate captions for each before concatenating them together. We use the following prompt to generate the captions for the first 50 frames: "Describe this video in detail, listing objects in each frame. Keep the descriptions concise." for TP2VR-objects. For any next sections, we use the prompt "Continue describing the video, listing objects in each frame. You are now at frame i, you have already described the previous i frames." For TP2VR-actions, we use the prompt "Describe this video in detail, listing actions and objects in each frame. Keep the descriptions concise." We set the `max_new_tokens` to 1024.

We use `CaptionSim` because the implementation of ELIOT is not publicly available. Our local implementation of ELIOT did not report good results (the video captions generated by (Tewel et al., 2022) did not include mentions of objects or actions).

## F QUERIES AND PROMPTS

We choose 15 temporal property templates for the experiments in Section 4.

The LogSTOP queries for these templates are as follows:

1. Eventually p1: $\diamond\, p1$
2. Always p1: $\square\, p1$
3. p1 Until p2: $p1\, \mathcal{U}\, p2$
4. Always p1 and Eventually p2: $\square\, p1 \wedge \diamond\, p2$
5. Always p1 or Eventually p2: $\square\, p1 \vee \diamond\, p2$
6. (Not p1) Until p2: $\neg p1\, \mathcal{U}\, p2$
7. p1 Until (Not p2): $p1\, \mathcal{U}\, \neg p2$
8. Always (p1 and p2): $\square(p1 \wedge p2)$
9. (p1 and p2) Until p3: $(p1 \wedge p2)\, \mathcal{U}\, p3$
10. p1 Until Always p2: $p1\, \mathcal{U}\, \square\, p2$
11. Eventually Always p1: $\diamond\, \square\, p1$
12. Always Eventually p1: $\square\, \diamond\, p1$
13. (Not p1) Until Eventually p2: $\neg p1\, \mathcal{U}\, \diamond\, p2$
14. (Not p1) Until Always p2: $\neg p1\, \mathcal{U}\, \square\, p2$
15. (p1 and p2) Until Eventually p3: $(p1 \wedge p2)\, \mathcal{U}\, \diamond\, p3$

NSVS-TL (Choi et al., 2024) uses the model checker STORM (Hensel et al., 2022) to verify if a given sequence satisfies a temporal property, where the temporal properties are represented in Probabilistic Computation Tree Logic (PCTL). In PCTL, the $F$, $G$ and $U$ operators represent the *Eventually*, *Always* and *Until* operators respectively. The $\sim$, & and $|$ operators represent the Boolean *negation*, *and*, and *or* operators respectively. The operator $P$ is used to indicate the ranges of probability of a given property being satisfied: for example, $P > 0.5[F\,\varphi]$ translates to "the probability of $\varphi$ eventually being satisfied is more than $0.5$".

The PCTL queries for the 15 temporal property templates are as follows:

1. Eventually p1: $P > 0.5\,[F\, "p1"\,]$
2. Always p1: $P > 0.5\,[G\, "p1"\,]$
3. p1 Until p2: $P > 0.5\,[\, "p1"\, U\, "p2"\,]$
4. Always p1 and Eventually p2: $P > 0.5\,[\, G\, "p1"\, \&\, F\, "p2"\,]$
5. Always p1 or Eventually p2: $P > 0.5\,[\, G\, "p1"\, |\, F\, "p2"\,]$
6. (Not p1) Until p2: $P > 0.5\,[\sim "p1"\, U\, "p2"\,]$
7. p1 Until (Not p2): $P > 0.5\,[\, "p1"\, U \sim "p2"\,]$
8. Always (p1 and p2): $P > 0.5\,[\, G\, "p1"\, \&\, G\, "p2"\,]$
9. (p1 and p2) Until p3: $P > 0.5\,[\, "p1"\, \&\, "p2"\, U\, "p3"\,]$
10. p1 Until Always p2: $P > 0.5\,[\, "p1"\, U\, G\, "p2"\,]$
11. Eventually Always p1: $P > 0.5\,[F\, G\, "p1"\,]$
12. Always Eventually p1: $P > 0.5\,[G\, F\, "p1"\,]$
13. (Not p1) Until Eventually p2: $P > 0.5\,[\sim "p1"\, U\, F\, "p2"\,]$
14. (Not p1) Until Always p2: $P > 0.5\,[\sim "p1"\, U\, G\, "p2"\,]$
15. (p1 and p2) Until Eventually p3: $P > 0.5\,[\, "p1"\, \&\, "p2"\, U\, F\, "p3"\,]$

The prompts for LVLMs for query matching on QMTP-video are as follows:

1. Eventually p1: "Is a $p1$ present in any frame of this video?"

2. Always p1: "Is a $p1$ present in all frames of this video?"

3. p1 Until p2: "Is a $p2$ present in any frame of this video and $p1$ present in all previous frames?"

4. Always p1 and Eventually p2: "Is a $p1$ present in all frames of this video and is a $p2$ present in any frame of this video?"

5. Always p1 or Eventually p2: "Is a $p1$ present in all frames of this video or is a $p2$ present in any frame of this video?"

6. (Not p1) Until p2: "Is a $p2$ present in any frame of this video and $p1$ absent in all previous frames?"

7. p1 Until (Not p2): "Is a $p2$ absent in any frame of this video and $p1$ present in all previous frames?"

8. Always (p1 and p2): "Are both $p1$ and $p2$ present in all frames of this video?"

9. (p1 and p2) Until p3: "Is a $p3$ present in any frame of this video and both $p1$ and $p2$ present in all previous frames?"

10. p1 Until Always p2: "Starting at some frame in this video, is a $p2$ present in all subsequent frames and $p1$ present in all previous frames?"

11. Eventually Always p1: "Starting at some frame in this video, is a $p1$ present in all subsequent frames?"

12. Always Eventually p1: "Starting at any frame in this video, is a $p1$ present in some subsequent frame?"

13. (Not p1) Until Eventually p2: "Starting at some frame in this video, is a $p2$ present in some subsequent frame and $p1$ absent in all previous frames?"

14. (Not p1) Until Always p2: "Starting at some frame in this video, is a $p2$ present in all subsequent frames and $p1$ absent in all previous frames?"

15. (p1 and p2) Until Eventually p3: "Starting at some frame in this video, is a $p3$ present in some subsequent frame and both $p1$ and $p2$ present in all previous frames?"

The prompts for LALMs for query matching on QMTP-speech are as follows:

1. Eventually p1: "Does the speaker's emotion sound $p1$ at any time?"

2. Always p1: "Does the speaker's emotion sound $p1$ at all times?"

3. p1 Until p2: "Does the speaker's emotion sound $p2$ at any time and $p1$ at all times until then?"

4. Always p1 and Eventually p2: "Does the speaker's emotion sound $p1$ at all times and $p2$ at any time?"

5. Always p1 or Eventually p2: "Does the speaker's emotion sound $p1$ at all times or $p2$ at any time?"

6. (Not p1) Until p2: "Does the speaker's emotion sound $p2$ at any time and not $p1$ at all times until then?"

7. p1 Until (Not p2): "Does the speaker's emotion sound not $p2$ at any time and $p1$ at all times until then?"

8. Always (p1 and p2): "Does the speaker's emotion sound both $p1$ and $p2$ at all times?"

9. (p1 and p2) Until p3: "Does the speaker's emotion sound $p3$ at any time and both $p1$ and $p2$ at all times until then?"

10. p1 Until Always p2: "Starting at some time in this audio clip, does the speaker's emotion sound $p2$ at all subsequent times and $p1$ at all previous times?"

11. Eventually Always p1: "Starting at some time in this audio clip, does the speaker's emotion sound $p1$ at all subsequent times?"

12. Always Eventually p1: "Starting at any time in this audio clip, does the speaker's emotion sound $p1$ at some subsequent time?"

13. (Not p1) Until Eventually p2: "Starting at some time in this audio clip, does the speaker's emotion sound $p2$ at some subsequent time and not $p1$ at all previous times?"

14. (Not p1) Until Always p2: "Starting at some time in this audio clip, does the speaker's emotion sound $p2$ at all subsequent times and not $p1$ at all previous times?"

15. (p1 and p2) Until Eventually p3: "Starting at some time in this audio clip, does the speaker's emotion sound $p3$ at some subsequent time and both $p1$ and $p2$ at all previous times?"

The queries for `mPLUG` and `CaptionSim` for retrieval on TP2VR-objects are as follows ($t_{lo} = 25$ and $t_{hi} = 50$ here):

1. Eventually p1: "A sequence of $t_{lo}$ to $t_{hi}$ frames where a $p1$ appears at some point."

2. Always p1: "A sequence of $t_{lo}$ to $t_{hi}$ frames where a $p1$ is always present."

3. p1 Until p2: "A sequence of $t_{lo}$ to $t_{hi}$ frames where a $p2$ is present at some point and a $p1$ is present in all frames before that."

4. Always p1 and Eventually p2: "A sequence of $t_{lo}$ to $t_{hi}$ frames where a $p1$ is always present and a $p2$ appears at some point."

5. Always p1 or Eventually p2: "A sequence of $t_{lo}$ to $t_{hi}$ frames where either a $p1$ is always present or a $p2$ appears at some point."

6. (Not p1) Until p2: "A sequence of $t_{lo}$ to $t_{hi}$ frames where a $p2$ is present at some point and a $p1$ is absent in all frames before that."

7. p1 Until (Not p2): "A sequence of $t_{lo}$ to $t_{hi}$ frames where a $p2$ is absent at some point and a $p1$ is present in all frames before that."

8. Always (p1 and p2): "A sequence of $t_{lo}$ to $t_{hi}$ frames where a $p1$ and a $p2$ are always present."

9. (p1 and p2) Until p3: "A sequence of $t_{lo}$ to $t_{hi}$ frames where a $p3$ is present at some point and both a $p1$ and a $p2$ are present in all frames before that."

10. p1 Until Always p2: "A sequence of $t_{lo}$ to $t_{hi}$ frames where starting at some point, a $p2$ is always present and a $p1$ is present in all frames before that."

11. Eventually Always p1: "A sequence of $t_{lo}$ to $t_{hi}$ frames where starting at some point, a $p1$ starts being always present."

12. Always Eventually p1: "A sequence of $t_{lo}$ to $t_{hi}$ frames where starting at any point, a $p1$ appears in some frames."

13. (Not p1) Until Eventually p2: "A sequence of $t_{lo}$ to $t_{hi}$ frames where starting at some point, a $p2$ appears in some frames and a $p1$ is absent in all frames before that."

14. (Not p1) Until Always p2: "A sequence of $t_{lo}$ to $t_{hi}$ frames where starting at some point, a $p2$ is always present and a $p1$ is absent in all frames before that."

15. (p1 and p2) Until Eventually p3: "A sequence of $t_{lo}$ to $t_{hi}$ frames where starting at some point, a $p3$ appears at some point and both a $p1$ and a $p2$ are present in all frames before that."

The queries for `mPLUG` and `CaptionSim` for retrieval on TP2VR-actions are as follows (note that $t_{lo} = t_{hi} = 50$ here):

1. Eventually p1: "A sequence of $t_{lo}$ frames where the action '$p1$' happens at some point."

2. Always p1: "A sequence of $t_{lo}$ frames where the action '$p1$' is always happening."

3. p1 Until p2: "A sequence of $t_{lo}$ frames where the action '$p2$' happens at some point and the action '$p1$' is happening in all frames before that."

4. Always p1 and Eventually p2: "A sequence of $t_{lo}$ frames where the action '$p1$' is always happening and the action '$p2$' happens at some point."

5. Always p1 or Eventually p2: "A sequence of $t_{lo}$ frames where either the action '$p1$' is always happening or the action '$p2$' happens at some point."

6. (Not p1) Until p2: "A sequence of $t_{lo}$ frames where the action '$p2$' happens at some point and the action '$p1$' is not happening in all frames before that."

7. p1 Until (Not p2): "A sequence of $t_{lo}$ frames where the action '$p2$' is not happening at some point and the action '$p1$' is happening in all frames before that."

8. Always (p1 and p2): "A sequence of $t_{lo}$ frames where the actions '$p1$' and '$p2$' are always happening."

9. (p1 and p2) Until p3: "A sequence of $t_{lo}$ frames where the action '$p3$' happens at some point and both the actions '$p1$' and '$p2$' are happening in all frames before that."

10. p1 Until Always p2: "A sequence of $t_{lo}$ frames where starting at some point, the action '$p2$' is always happening and the action '$p1$' is happening in all frames before that."

11. Eventually Always p1: "A sequence of $t_{lo}$ frames where starting at some point, the action '$p1$' is always happening."

12. Always Eventually p1: "A sequence of $t_{lo}$ frames where starting at any point, the action '$p1$' happens in some frames."

13. (Not p1) Until Eventually p2: "A sequence of $t_{lo}$ frames where starting at some point, the action '$p2$' happens in some frames and the action '$p1$' is not happening in all frames before that."

14. (Not p1) Until Always p2: "A sequence of $t_{lo}$ frames where starting at some point, the action '$p2$' is always happening and the action '$p1$' is not happening in all frames before that."

15. (p1 and p2) Until Eventually p3: "A sequence of $t_{lo}$ frames where starting at some point, the action '$p3$' happens at some point and both the actions '$p1$' and '$p2$' are happening in all frames before that."

## G  OTHER EXPERIMENT DETAILS

**Compute Resources.** All experiments were run on a shared cluster with the following GPUs: eight NVIDIA A100 PCIe (80GB RAM each) and eight NVIDIA RTX A6000 (48GB RAM each).

## H  EXAMPLES

Figure 5 presents examples of ranking using the three methods. In the first example, the video captions used by `CaptionSim` do not include smaller objects that might be relevant to the query ("car" in this example). In the second example, the two actions are mentioned in the caption – the video is ranked lower than other videos with more mentions of the actions ("stand" and "hand clap" in this case). This demonstrates that while caption-based methods outperform joint model embeddings (`mPLUG`), they rely on semantic similarity between captions and text to determine relevance, which might not be sufficient for effective retrieval with temporal queries.

## I  ABLATIONS

### I.1  ADAPTIVE THRESHOLD VS. CONSTANT THRESHOLD

Figure 6 presents a comparison of the adaptive threshold and the constant $\log 0.5$ threshold for all temporal property templates, using LogSTOPs for matching and non-matching sequences from the QMTP-video (detections using YOLOv8). For all properties, the adaptive threshold accepts at least as many matching sequences as the constant threshold.

### I.2  CHOICE OF BASE THRESHOLDS

In Section 3.1, we define an adaptive threshold over $\log 0.5$ for query matching with LogSTOP. We could choose a different base threshold value in the range $\{\log x : x \in [0, 1]\}$. In Figure 7, we plot

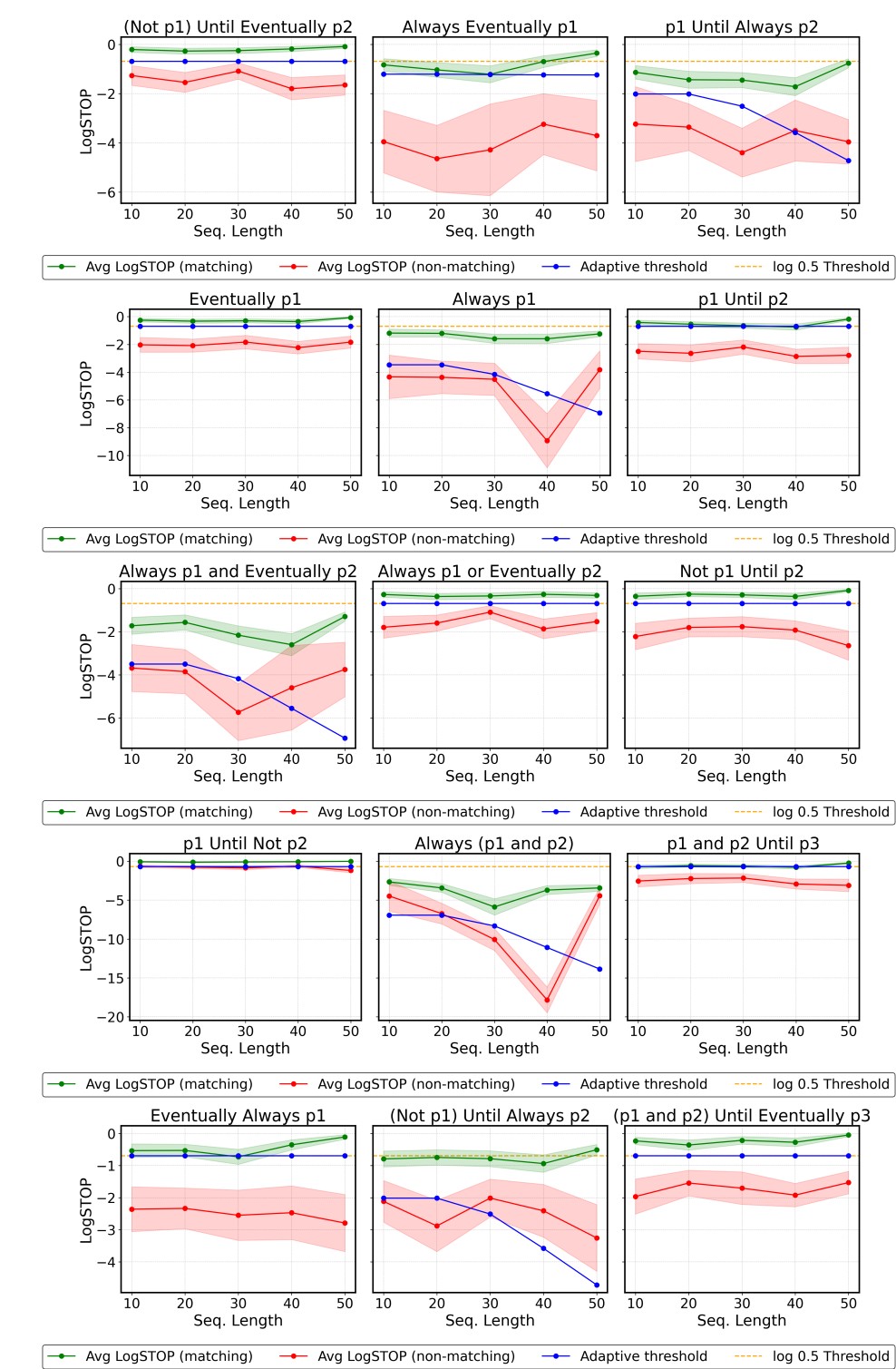

Figure 6: The adaptive threshold accepts more matching sequences than the constant $\log 0.5$ threshold. LogSTOPs with YOLOv8 (mean with 95% CI) are shown for sequences from QMTP-video.

the balanced accuracies of query matching on sequences of length 50 from QMTP-video against a sweep over $0.1$ increments of this base threshold value for YOLOv8 and GroundingDINO. For a given base threshold value, the adaptive threshold is calculated by replacing $\log 0.5$ with the new value. We find that LogSTOP reports the highest accuracies with both models using the $\log 0.5$

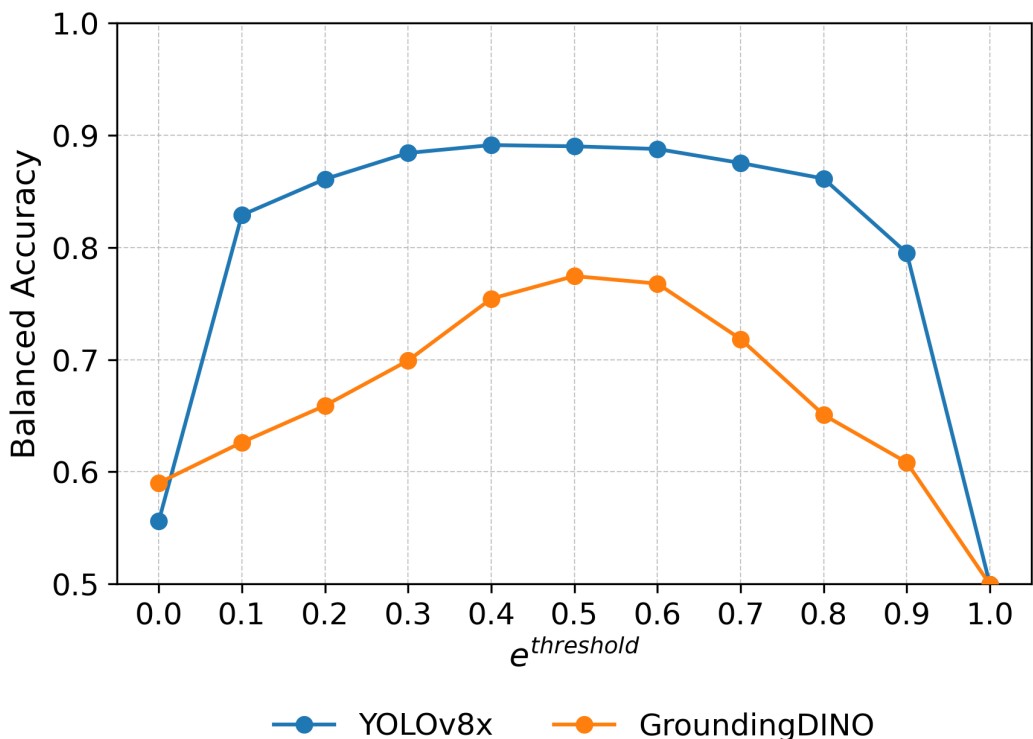

Figure 7: LogSTOP's performance on query matching depends on the value of the base threshold. The best performance is reported by thresholds around $\log 0.5$. The results are on sequences on length 50 from the QMTP-Video dataset.

threshold. Interestingly, we find that there is a range of thresholds around $\log 0.5$ where the accuracies are high. For YOLOv8, base threshold values in the range $[\log 0.2, \log 0.8]$ report accuracies over 85%. GroundingDINO, on the other hand, reports accuracies over 75% with a narrower range of $[\log 0.4, \log 0.6]$.

### I.3 CHOICE OF SMOOTHING WINDOW

LogSTOP uses a downsampling-smoothing window $w$ and the choice of this parameter impacts performance on downstream tasks. We consider sequences of length 20 from QMTP-Video and report the balanced accuracies and accuracies on matching sequences for different values of $w$ in Table 2. We find that the balanced accuracies peak at lower values of $w \in \{4, 5\}$ and are lower with no smoothing ($w = 1$) or extreme smoothing ($w = 20$). Interestingly, there are intuitive trends for different temporal properties. For properties such as "Always p1", "Always (p1 and p2)", and "(Not p1) Until Always p2", the predictions for p1 and p2 are expected to be high/low for longer contiguous subsequences and hence query matching performance steadily improves as the value of the smoothing window $w$ increases. On the other hand, for properties such as "Eventually p1" and "Always Eventually p1", the predictions for p1 can be high/low for shorter contiguous subsequences (even a single high/low prediction is sufficient to match / not match, respectively), and hence these properties benefit from lower values of $w$. Finally, properties with a mix of these prediction patterns benefit from moderate values of $w$; for instance, the property "(Not p1) Until Eventually p2" can be matched by sequences with a long contiguous subsequence of low p1 predictions and a single timestep with a high prediction for p2.

| Query | $w = 1$ | $w = 2$ | $w = 4$ | $w = 5$ | $w = 10$ | $w = 20$ |
|---|---|---|---|---|---|---|
| Eventually p1 | 0.82 | 0.82 | 0.82 | **0.83** | 0.77 | 0.7 |
| Always p1 | 0.75 | 0.76 | 0.8 | 0.8 | **0.82** | **0.82** |
| p1 Until p2 | 0.79 | 0.81 | 0.81 | **0.82** | 0.79 | 0.75 |
| Always p1 and Eventually p2 | 0.7 | **0.72** | 0.71 | 0.71 | 0.71 | **0.72** |
| Always p1 or Eventually p2 | 0.75 | 0.72 | 0.77 | 0.75 | **0.79** | 0.69 |
| Not p1 Until p2 | 0.83 | 0.86 | **0.88** | **0.88** | 0.86 | 0.77 |
| p1 Until Not p2 | 0.73 | 0.7 | 0.75 | 0.74 | 0.77 | **0.82** |
| Always (p1 and p2) | 0.61 | 0.65 | 0.66 | 0.69 | **0.72** | **0.72** |
| p1 and p2 Until p3 | **0.83** | 0.82 | 0.78 | 0.79 | 0.73 | 0.7 |
| p1 Until Always p2 | 0.68 | 0.7 | 0.71 | 0.71 | **0.73** | 0.71 |
| Eventually Always p1 | 0.75 | **0.8** | **0.8** | 0.79 | 0.78 | 0.73 |
| Always Eventually p1 | **0.77** | **0.77** | 0.76 | 0.76 | 0.75 | 0.76 |
| (Not p1) Until Eventually p2 | 0.79 | 0.81 | **0.84** | 0.83 | 0.78 | 0.71 |
| (Not p1) Until Always p2 | 0.71 | 0.79 | 0.77 | 0.76 | **0.82** | 0.8 |
| (p1 and p2) Until Eventually p3 | 0.77 | **0.8** | 0.77 | 0.78 | 0.78 | 0.69 |
| Overall | 0.75 | 0.77 | **0.78** | **0.78** | 0.77 | 0.74 |

Table 2: The value of the downsampling-smoothing window $w$ affects the performance of LogSTOP on query matching. Temporal properties exhibit trends depending on whether sequences require long contiguous subsequences to determine matching. The results are using LogSTOP with YOLOv8x on videos with 20 frames from the QMTP-Video dataset. The best performing method is highlighted in **bold** and the second best is underlined.

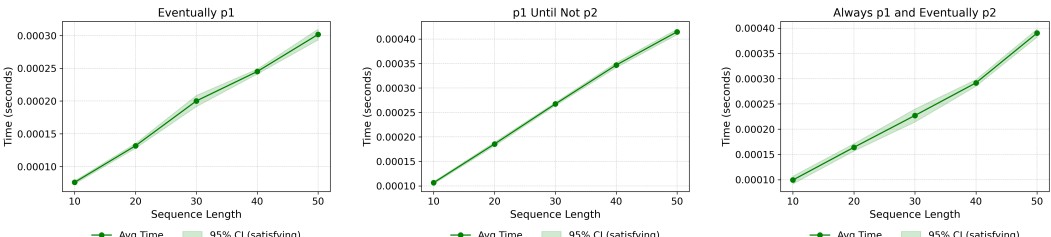

Figure 8: Time taken for query matching with LogSTOP scales linearly with time for temporal properties of varying lengths. The results are using YOLOv8x on the QMTP-Video benchmark. Note that this does not include the time taken by the object detector.

## I.4 EFFICIENCY OF LOGSTOP

Finally, we analyze the practical efficiency of LogSTOP. Algorithm 1 has a computational complexity of $\mathcal{O}(|\varphi| \cdot T)$, where $|\varphi|$ is the length of the temporal property and $T$ is the length of the sequence. In Figure 8, we plot the time taken by LogSTOP against the length of the sequences from QMTP-Video. The time taken to compute LogSTOPs scales roughly linearly with the length of the sequence for a given temporal property.

## J    DETAILED RESULTS

### J.1    RANKED RETRIEVAL

In Section 4, we present the results for ranked retrieval with different methods. Table 3 reports these results (with additional metrics, including Precision@5 and Precision@10).

Table 3: Retrieval results on the TP2VR datasets.

| Method | P@1↑ | P@5↑ | P@10↑ | P@r↑ | mAP↑ | R@r↑ | MnR↓ | MdR↓ |
|---|---|---|---|---|---|---|---|---|
| *TP2VR-objects (mean r = 163)* | | | | | | | | |
| *mPLUG* | 0.10 | 0.22 | 0.33 | 0.35 | 0.34 | 0.35 | 8.1 | 7.0 |
| *CaptionSim* | 0.50 | 0.64 | 0.57 | 0.35 | 0.36 | 0.35 | 3.1 | 2.0 |
| *LogSTOP (GroundingDINO)* | 0.79 | 0.77 | 0.79 | 0.59 | 0.64 | 0.59 | 2.0 | 1.0 |
| *TP2VR-actions (mean r = 21)* | | | | | | | | |
| *mPLUG* | 0.07 | 0.05 | 0.06 | 0.05 | 0.06 | 0.05 | 60.7 | 28.5 |
| *CaptionSim* | 0.23 | 0.15 | 0.17 | 0.11 | 0.09 | 0.11 | 20.4 | 5.0 |
| *LogSTOP (SlowR50)* | 0.47 | 0.38 | 0.37 | 0.27 | 0.28 | 0.27 | 7.9 | 2.0 |

## J.2 QUERY MATCHING

In Section 4, we present the query matching results for the QMTP datasets. Table 4 presents the results (balanced accuracies) aggregated by category. Table 5 Table 6 report the results for the 15 temporal property templates for QMTP-video and QMTP-speech respectively.

In Table 7, we provide results on query matching with a larger LVLM, namely `InternVL2-26B`. With access to 5 A100 GPUs, the model could support videos with up to 30 frames with 16-bit precision and we hence restrict comparison with other methods to this subset. Despite being significantly larger than the other 7B VLMs and local property predictors used by LogSTOP, `InternVL2-26B` only outperforms `LongVA-7B` by 1% in terms of balanced accuracy. LogSTOP with Grounding DINO and YOLOv8x outperforms `InternVL2-26B` by 3% and 13% respectively.

Table 4: Average balanced accuracy for each temporal property category (columns) and method (rows) on the QMTP-video and QMTP-speech datasets. Detailed results per category and sequence length are in Appendix J.

| Method | Simple | Bool. over Temp. | Temp. over Bool. | Temp. over Temp. | Mixed | Overall |
|---|---|---|---|---|---|---|
| *QMTP-video* | | | | | | |
| *NSVS-TL* | 0.67 | 0.63 | 0.51 | 0.64 | 0.50 | 0.58 |
| *Video-LLaVA-7B* | 0.50 | 0.50 | 0.50 | 0.50 | 0.50 | 0.50 |
| *LongVA-7B* | 0.68 | 0.63 | 0.62 | 0.63 | 0.61 | 0.63 |
| *LogSTOP (OWLv2)* | 0.61 | 0.60 | 0.58 | 0.58 | 0.63 | 0.60 |
| *LogSTOP (GroundingDINO)* | 0.70 | 0.63 | 0.69 | 0.70 | 0.64 | 0.68 |
| *LogSTOP (YOLOv8)* | 0.82 | 0.75 | 0.78 | 0.77 | 0.81 | 0.79 |
| *QMTP-speech* | | | | | | |
| *Qwen-Audio-Chat* | 0.70 | 0.58 | 0.71 | 0.66 | 0.64 | 0.68 |
| *Qwen2-Audio-7B-Instruct* | 0.64 | 0.64 | 0.65 | 0.58 | 0.56 | 0.63 |
| *LogSTOP (HuBERT)* | 0.90 | 0.77 | 0.80 | 0.83 | 0.79 | 0.84 |

Table 5: LogSTOP reports the best overall balanced accuracy on the QMTP-video dataset, outperforming LVLMs and NSVS-TL. Moreover, it reports the best performance on all queries. The dataset contains 3750 matching and 3718 non-matching sequences of lengths $\{10, 20, 30, 40, 50\}$. The best performing method is highlighted in **bold** and the second best is underlined.

| Query | NSVS-TL | Video-LLaVA-7B | LongVA-7B | | LogSTOP | |
| --- | --- | --- | --- | --- | --- | --- |
| | | | | OWLv2 | GroundingDINO | YOLOv8 |
| Eventually p1 | 0.62 | 0.5 | 0.71 | 0.65 | 0.62 | **0.84** |
| Always p1 | 0.68 | 0.51 | 0.72 | 0.54 | 0.76 | **0.81** |
| p1 Until p2 | 0.7 | 0.5 | 0.61 | 0.63 | 0.73 | **0.8** |
| Always p1 and Eventually p2 | 0.55 | 0.5 | 0.65 | 0.51 | 0.65 | **0.72** |
| Always p1 or Eventually p2 | 0.71 | 0.5 | 0.61 | 0.68 | 0.61 | **0.78** |
| Not p1 Until p2 | 0.5 | 0.5 | 0.68 | 0.71 | 0.77 | **0.85** |
| p1 Until Not p2 | 0.5 | 0.5 | 0.64 | 0.53 | 0.66 | **0.78** |
| Always (p1 and p2) | 0.55 | 0.51 | 0.62 | 0.51 | 0.65 | **0.73** |
| p1 and p2 Until p3 | 0.5 | 0.5 | 0.55 | 0.56 | 0.71 | **0.78** |
| p1 Until Always p2 | 0.64 | 0.5 | 0.57 | 0.51 | 0.62 | **0.75** |
| Eventually Always p1 | 0.64 | 0.5 | 0.62 | 0.7 | 0.76 | **0.79** |
| Always Eventually p1 | 0.64 | 0.5 | 0.69 | 0.55 | 0.72 | **0.76** |
| (Not p1) Until Eventually p2 | 0.5 | 0.5 | 0.61 | 0.59 | 0.59 | **0.81** |
| (Not p1) Until Always p2 | 0.5 | 0.5 | 0.68 | 0.66 | 0.71 | **0.78** |
| (p1 and p2) Until Eventually p3 | 0.5 | 0.5 | 0.55 | 0.65 | 0.62 | **0.84** |
| Overall | 0.58 | 0.5 | 0.63 | 0.6 | 0.68 | **0.79** |

Table 6: LogSTOP with HuBERT outperforms LALMs on the QMTP-speech dataset, both overall and on each query. We report balanced accuracies on 1650 matching and 1650 non-matching sequences of lengths 5-30. The best performing method is highlighted in **bold** and the second best is underlined.

| Query | Qwen-Audio-Chat | Qwen2-Audio-7B-Instruct | LogSTOP (HuBERT) |
| --- | --- | --- | --- |
| Eventually p1 | 0.75 | 0.72 | **0.94** |
| Always p1 | 0.67 | 0.66 | **0.87** |
| p1 Until p2 | 0.69 | 0.53 | **0.9** |
| Always p1 or Eventually p2 | 0.58 | 0.64 | **0.77** |
| Not p1 Until p2 | 0.77 | 0.72 | **0.8** |
| p1 Until Not p2 | 0.64 | 0.7 | **0.8** |
| p1 Until Always p2 | 0.52 | 0.53 | **0.82** |
| Eventually Always p1 | 0.75 | 0.74 | **0.92** |
| Always Eventually p1 | 0.69 | 0.5 | **0.76** |
| (Not p1) Until Eventually p2 | 0.68 | 0.55 | **0.76** |
| (Not p1) Until Always p2 | 0.59 | 0.49 | **0.79** |
| Overall | 0.68 | 0.63 | **0.84** |

Table 7: LogSTOP outperforms larger LVLMs such as `InternVL2-26B` on query matching by upto 13% in terms of balanced accuracy. This comparison only focuses on a subset of QMTP-Video, with videos containing up to 30 frames. The best performing method is highlighted in **bold** and the second best is underlined.

| Query | NSVS-TL | Video-LLaVA-7B | LongVA-7B | InternVL2-26B | LogSTOP | | |
| --- | --- | --- | --- | --- | --- | --- | --- |
| | | | | | OWLv2 | GroundingDINO | YOLOv8 |
| Eventually p1 | 0.6 | 0.5 | 0.71 | 0.76 | 0.71 | 0.65 | **0.82** |
| Always p1 | 0.66 | 0.51 | 0.72 | 0.74 | 0.54 | 0.72 | **0.77** |
| p1 Until p2 | 0.68 | 0.5 | 0.64 | 0.58 | 0.63 | 0.74 | **0.79** |
| Always p1 and Eventually p2 | 0.54 | 0.5 | 0.66 | 0.62 | 0.51 | 0.63 | **0.7** |
| Always p1 or Eventually p2 | 0.69 | 0.5 | 0.6 | 0.57 | 0.67 | 0.61 | **0.75** |
| Not p1 Until p2 | 0.5 | 0.5 | 0.67 | 0.64 | 0.68 | 0.73 | **0.84** |
| p1 Until Not p2 | 0.5 | 0.5 | 0.61 | 0.7 | 0.56 | 0.63 | **0.78** |
| Always (p1 and p2) | 0.54 | 0.51 | 0.61 | 0.62 | 0.51 | 0.6 | **0.68** |
| p1 and p2 Until p3 | 0.5 | 0.5 | 0.56 | 0.56 | 0.57 | 0.67 | **0.76** |
| p1 Until Always p2 | 0.63 | 0.5 | 0.56 | 0.51 | 0.51 | 0.6 | **0.72** |
| Eventually Always p1 | 0.6 | 0.5 | 0.56 | 0.64 | 0.67 | 0.7 | **0.75** |
| Always Eventually p1 | 0.6 | 0.5 | 0.66 | **0.74** | 0.54 | 0.66 | 0.72 |
| (Not p1) Until Eventually p2 | 0.5 | 0.5 | 0.6 | 0.64 | 0.61 | 0.59 | **0.77** |
| (Not p1) Until Always p2 | 0.5 | 0.5 | 0.65 | 0.56 | 0.63 | 0.66 | **0.75** |
| (p1 and p2) Until Eventually p3 | 0.5 | 0.5 | 0.55 | 0.59 | 0.71 | 0.64 | **0.82** |
| Overall | 0.57 | 0.5 | 0.62 | 0.63 | 0.6 | 0.66 | 0.76 |

## K   LARGE LANGUAGE MODELS USAGE STATEMENT

During the preparation of this work, we used a large language model for checking grammar and suggesting LaTeX commands.

