# OpenReview forum: "LogSTOP: Temporal Scores over Prediction Sequences for Matching and Retrieval"
_ICLR.cc/2026/Conference — Submitted to ICLR 2026_

### Official Review · Reviewer_E262 · 2025-10-30

**Soundness:** 3
**Presentation:** 3
**Contribution:** 3
**Rating:** 6
**Confidence:** 3

**Summary:**

This paper studies how to lift local property detection scores in videos or audio to sequence-level temporal property scores, for example using YOLO to detect objects in video frames or HuBERT to detect emotions in audio clips, with confidence scores in [0,1]. The authors formalize this task as the “Scoring for TempOral Properties (STOPs)” problem, aiming to map these scores to temporal properties expressed in Linear Temporal Logic (LTL) despite potential noise in local predictions. To address this, the paper proposes a novel scoring function, LogSTOP, which efficiently computes sequence-level temporal property scores and handles temporal patterns such as “eventually” or “until.” Overall, this work presents a theoretically sound and computationally efficient approach that effectively lifts local prediction scores to temporal logic properties and demonstrates significant advantages across multiple downstream tasks.

**Strengths:**

1.One of the main contributions of this paper is the clear formalization of an important and challenging problem: STOPs (Scores for Temporal Properties), which involves lifting frame-level scores from potentially noisy local detectors to sequence-level scores for complex LTL properties.
2.The LogSTOP algorithm is designed to be efficient and scalable, making it well-suited for practical applications such as large-scale retrieval.
3.Although its theoretical assumptions (e.g., independence) have limitations, LogSTOP is highly effective in practice and is specifically designed to handle noisy real-world data.

**Weaknesses:**

1.The paper uses $P(\phi_1 \wedge \phi_2) \approx P(\phi_1) P(\phi_2) $(addition in log space) as the conjunction combination rule, which is equivalent to a strong independence assumption: the sub-events are independent and their probabilities are well-calibrated. However, in video or audio, the same attribute is often strongly correlated across adjacent time segments—e.g., an object persists across frames—and different attributes are also frequently correlated, such as a person appearing together with walking actions. The independence assumption therefore generally does not hold. Although the paper acknowledges that this assumption often fails, it represents a core logical inconsistency, as treating LogSTOP as a probabilistic model relies on a fundamentally incorrect assumption. Consequently, the effectiveness of LogSTOP may not stem from the paper’s central conceptual framework.
2.A question regarding the smoothing window is that it modifies the semantics of temporal logic (LTL). In LogSTOP, the “Next” operator means “φ holds in the next w window” rather than the standard LTL meaning of “φ holds at the next time point t+1.” This fundamentally changes the evaluated logical properties. An “eventually” property under the t+1 semantics is completely different from under the t+w semantics. Did the authors provide any explanation for this?
3.A question regarding the experimental section is the inconsistency between evaluation semantics and algorithm semantics. The ground-truth labels for QMTP and TP2VR are generated based on “standard LTL (t+1),” whereas LogSTOP actually computes “coarse-grained LTL (t+w)”. This means the model is evaluated under a different semantic framework, and the authors should provide an explanation for this.

**Questions:**

1.While the paper notes that the assumption often does not hold in video or audio sequences, the authors could further analyze its impact on the theoretical interpretation of LogSTOP. They could include experiments or analyses to show why the model still performs well empirically even when the independence assumption is violated, and discuss that the performance may stem more from empirical properties than from strict probabilistic modeling.
2.Explain how the use of a window w in LogSTOP changes the semantics of the “Next” operator compared to standard LTL (t+1), and discuss the potential bias or limitations this semantic modification introduces in evaluating temporal properties.
3.Clarify that the ground-truth labels for QMTP and TP2VR are generated based on standard LTL, whereas LogSTOP computes coarse-grained LTL , which may lead to an inconsistency in evaluation semantics. It is recommended to provide qualitative analysis or ablation experiments to assess the effect of this semantic difference on results, or to justify why comparisons remain fair in practice.

---

> ### Author Response · Authors · 2025-11-20
> **Response to reviewer E262 (1/2)**
>
> We would like to thank the reviewer for appreciating our formalization of the STOP problem and the practical applicability of LogSTOP. We also appreciate the insightful comments on the semantics and assumptions, and we present answers to the reviewer’s questions here:
>
> **Q1: Other empirical properties that could explain LogSTOP’s effectiveness when the assumptions are violated**
>
> This is a very interesting comment. We think that the effectiveness of LogSTOP on the downstream tasks can be attributed to the following:
>
> - The downstream tasks, namely query matching and ranked retrieval, do not require the scores to be true probabilities. Concretely, the scores only need to reflect the right ranking order: sequences that match or are more relevant to the temporal property need to be scored higher than sequences that do not match or are less relevant to the temporal property.
> - In order to understand this further, we analyze how dependencies between local properties in a frame affect LogSTOP scores. We acknowledge that this is only a subset of independence assumptions (we do not look at temporal dependencies) but we choose this setting because it is easier to analyze statistical dependencies between local properties in individual frames. Concretely, we focus on the temporal property template “Always (p1 and p2)” and evaluate how LogSTOP’s accuracy on query matching changes for different choices of p1 and p2.
>
> *Setup:* We focus on local properties from the RealTLV dataset, i.e., 6 object classes (car, person, bus, truck, bicycle, and motorcycle). We assess the pair-wise statistical dependence between these classes in a frame using the Chi-square test of independence and mutual information over the frame-level ground truth annotations. We find that the two metrics agree on the order of independence. Simultaneously, we compute the class-wise accuracy of YOLOv8 on the dataset and run LogSTOP with YOLOv8 for query matching with the property “Always (p1 and p2)”, for different choices of p1 and p2.
>
> *Results:* Interestingly, we find that LogSTOP performs better on pairs where the class-wise accuracy of YOLOv8 is higher, irrespective of how statistically dependent the two classes are. Concretely, the accuracy of object detection using YOLOv8 follows the order car (85%) > person (59%) > truck (48%) > bus (45%). The order of F1 scores for query matching is (car, person) (0.64) > (car, truck) (0.57) > (person, truck) (0.33) > (bus, truck) (0.15), which roughly follows the order of decreasing object detection accuracy. This is different from the order of dependence from the statistical tests (increasing p-values from the Chi-square test of independence and decreasing mutual information): (bus, truck) > (car, person) > (person, truck) > (car, truck).
>
> We also find that the frequency of classes in the dataset is highly correlated with the object detection accuracy, indicating that samples in the benchmark are more likely to include local properties with higher detection accuracies, in turn leading to better LogSTOPs. The high aggregate performance of LogSTOP on query matching can then potentially be explained in terms of the representation of these classes in the QMTP-Video benchmark.
>
> We hope that these insights provide a clear and intuitive guideline for how and when LogSTOP is expected to perform well in practice – concretely, on temporal properties over local properties with more accurate predictors. We would also appreciate the reviewer’s suggestions on other tests / analyses that we could include here.

---

> > ### Author Response · Authors · 2025-11-20
> > **Response to reviewer E262 continued (2/2)**
> >
> > **Q2: Explain how the use of a window w in LogSTOP changes the semantics of the “Next” operator compared to standard LTL (t+1), and discuss the potential bias or limitations this semantic modification introduces in evaluating temporal properties.**
> >
> > We discuss the changes to the semantics of the Next operator with smoothing in Appendix A. Concretely, for any sequence and temporal property, the LogSTOPs could change drastically with the smoothing window used. The smoothing window is hence an important parameter where higher values result in “more compressed” prediction sequences – this is akin to the standard t+1 semantics on the sequence of average-aggregated blocks of width w.
> >
> > *Ablation study on the smoothing window:* We present an ablation study on the impact of the smoothing window w on LogSTOP’s query matching performance in Appendix I.3. Concretely, we consider sequences of length 20 from our QMTP-Video benchmark and evaluate LogSTOP (YOLOv8x) with smoothing windows in [1, 2, 4, 5, 10, 20]. We find that the balanced accuracies (over matching and non-matching sequences) are highest with w=4 and w=5, and lowest with w=1 and w=20 (Table 2). In general, low accuracy with w=1 corroborates our hypothesis of how noisy local predictions can affect the score adversely, while low accuracy with w=20 highlights the loss of information as the sequence is compressed to a single prediction point. We also find intuitive temporal property specific trends: properties that require homogeneous predictions (“a car in all frames”, for example) benefit from increased smoothing, while properties with infrequent deciding local predictions (“a car in some frames”) benefit from reduced smoothing. Most properties involve a combination of these prediction trends and hence benefit from moderate values of w.
> >
> > **Q3: Clarify that the ground-truth labels for QMTP and TP2VR are generated based on standard LTL, whereas LogSTOP computes coarse-grained LTL , which may lead to an inconsistency in evaluation semantics. It is recommended to provide qualitative analysis or ablation experiments to assess the effect of this semantic difference on results, or to justify why comparisons remain fair in practice.**
> >
> > Our discussion on benchmark creation in Appendix C states that we use the standard LTL semantics for computing ground truth labels for the two benchmarks. Our evaluation is consistent for the following reasons:
> > - Firstly, the standard t+1 LTL semantics is the ground truth for all tasks in our benchmark. The window is a property of the proposed method (LogSTOP) and hence does not (and should not) affect the ground truth labels. Since the window is a hyperparameter, it should be carefully chosen to reduce the impact of noisy local predictions while retaining enough temporal structure. Our results on query matching and the ablation study in Appendix I.3 (discussed in detail in response to Q2) suggest that lower moderate values of w are suitable to balance these extremes for most temporal properties.
> > - Secondly, the evaluation for the downstream tasks is consistent in that (1) the adaptive threshold used for matching uses the same smoothing window as the sequence being evaluated, and (2) the same smoothing window is used for computing LogSTOPs for all sequences when ranking for retrieval. We acknowledge that this may have been inconsistent if different smoothing windows were used for comparing sequences.
> > - Thirdly, we only use the smoothing window as a hyper parameter with LogSTOP and hence, this parameter does not affect the evaluation of other baselines.
> >
> > We would be happy to discuss any other sources of inconsistencies in the evaluation with the reviewer.

---

### Official Review · Reviewer_4Lc6 · 2025-10-31

**Soundness:** 3
**Presentation:** 3
**Contribution:** 3
**Rating:** 4
**Confidence:** 3

**Summary:**

This paper propose a scoring function  that can efficiently compute scores for temporal properties represented in Linear Temporal Logic. This module enables reasoning over sequences of predictions. To support the experiment, this paper also built two bench marks. Experiments show they achieve better results.

**Strengths:**

1, The introduction of the two benchmarks are important for this community
2, The idea on sequence scoring is easy and plug-and-play.
3, The writing is good in the motivation part.

**Weaknesses:**

1, The experiments presented is not in detail. Its difficult to read their specific numbers and for future citation and comparions.
2, The quality and collection details of the two tasks are not discussed. This is very important in evaluating the significance of two bench marks.
3, The related work is very brief. Lack discussion on weakness of existing papers and fails to build relations between this work and previous works.
4, Line 52 is difficult to understand
5, Notations in Algorithm 1 is difficult to understand, some of them came with no context or definition.

I consider to improve the score if the significance of benchmarks are further verified.

**Questions:**

1, Would LogStop work beyond video and speech domains?
2，The logic seems to be manually defined and domain-specific. How generalizable is this approach to future or more complex tasks where temporal relations may not follow fixed rule templates
3, please see weakness related to the benchmarks.

---

> ### Author Response · Authors · 2025-11-20
> **Response to reviewer 4Lc6 (1/2)**
>
> We appreciate the reviewer’s positive remarks on the importance of the benchmarks and the simplicity of our approach. We would like to first answer the questions from the reviewer, before providing more details on the experiments and benchmarks.
>
> **Q1: Would LogStop work beyond video and speech domains?**
>
> LogSTOP is domain-agnostic and can be used to score arbitrary temporal properties in Linear Temporal Logic as long as local properties can be defined with associated predictors. For instance, given text sentiment predictors for posts on a social media platform, LogSTOP can be used to check if a thread of posts on a user’s timeline is “eventually positive” or “positive until always neutral”. Similarly, LogSTOP can be used in multi-modal domains – given a video with the associated audio, LogSTOP can score it against properties such as  “a person is walking and speaking in English until they eventually sit down and stop speaking”. The local property predictors required to score this property would be an object detector (to detect a person), an action recognition model (to detect the walking, speaking, and sitting actions), and a speech detector (to detect English speech and silence).
>
>
> **Q2: The logic seems to be manually defined and domain-specific. How generalizable is this approach to future or more complex tasks where temporal relations may not follow fixed rule templates?**
>
> We use Linear Temporal Logic (LTL) to represent temporal properties over arbitrary local properties from arbitrary domains. While our experiments and examples focus on objects and actions in videos and emotions in speech, we can represent very complex temporal properties by choosing the right abstraction for local properties (as in the examples presented in response to Q1). LTL and other similar temporal logics have been extensively used to write specifications: some recent examples include LTL for specifying tasks for RL agents [1], a variant of LTL for  natural language video captions [2], and PCTL for video generation [3].
>
>
> **W1: More details on the experiments and numbers**
>
> Due to space constraints, we present the key results as bar charts in Section 4 and provide the detailed experiment setup and results in the Appendix. We would like to point the reviewer to Appendix J for detailed results. The specific numbers for ranked retrieval, corresponding to Figure 4 in the main text, are presented in Table 3. The category-aggregate numbers  for query matching corresponding to Figure 2 in the main text are provided in Table 4, with detailed results for QMTP-Video and QMTP-speech in Tables 5 and 6 respectively. The details of all methods for query matching and ranked retrieval are presented in Appendix D and E respectively. All the prompts and temporal queries are listed in Appendix F and the computational resources used to run the experiments are mentioned in Appendix G.
>
>
> *References*
>
> [1] Jackermeier, M., & Abate, A. (2024). Deepltl: Learning to efficiently satisfy complex ltl specifications for multi-task rl. ICLR 2025.
>
> [2] Huang, J., Li, Z., Naik, M., & Lim, S. N. (2023). Laser: A neuro-symbolic framework for learning spatial-temporal scene graphs with weak supervision. ICLR 2025.
>
> [3] ​​Sharan, S. P., Choi, M., Shah, S., Goel, H., Omama, M., & Chinchali, S. (2025). Neuro-symbolic evaluation of text-to-video models using formal verification. In Proceedings of the Computer Vision and Pattern Recognition Conference (pp. 8395-8405).

---

> > ### Author Response · Authors · 2025-11-20
> > **Response to reviewer 4Lc6 continued (2/2)**
> >
> > **W2: The quality and collection details of the two tasks are not discussed. This is very important in evaluating the significance of two benchmarks.**
> >
> > We would like to point the reviewer to Appendix C for details about the data sources and the benchmark creation pipeline. We present a summary of the details here:
> >
> > *Quality and significance of Data sources:* Our benchmarks are created using existing datasets – Real-TLV which includes videos from the NuScenes and Waymo driving datasets, IEMOCAP which includes audio clips of conversations between 2 speakers, and AVA which consists of 15-min video clips from YouTube. We choose these datasets for the following reasons:
> >
> > - These datasets have extensively been used by other works on object detection, emotion recognition, and action recognition, respectively, confirming the credibility of the datasets.
> > - The datasets contain samples from two distinct domains (video and speech), with a diverse array of local properties (objects, actions, emotions).
> > - The datasets provide frame-level annotations for objects (car, pedestrian, etc.), emotions (happy, sad, etc.), and actions (standing, walking, etc.) respectively. Our benchmark creation pipeline can compute ground truth labels for arbitrary temporal properties using standard LTL semantics over these frame-level annotations.
> > - We can evaluate several temporal properties of practical interest on these datasets. Concretely, surveillance queries such as “a bicycle eventually disappears”, movie searches with scenes where “a person is talking on the phone until they start driving”, and emotion patterns such as “the speaker eventually sounds frustrated” can be written on samples from the RealTLV, AVA, and IEMOCAP datasets respectively.
> >
> > *Benchmark creation pipeline:* We provide a fully automated benchmark creation pipeline that can generate samples for any temporal property, given datasets with fully-annotated local properties. We use this pipeline on the three datasets mentioned above with 15 temporal property templates mentioned in Section 4. Concretely, given a target sequence and a temporal property, we use the standard LTL semantics to compute ground truth binary labels. We will open-source this benchmark creation pipeline for the two tasks, namely query matching and ranked retrieval, and hope that future work can use it with datasets from other domains and with other temporal property templates.
> >
> > *Statistics:* As we discuss in Section 4.1, the query matching benchmark (QMTP) consists of 7468 video samples (from the Real-TLV dataset) and 3300 speech samples (from the IEMOCAP dataset), with binary labels. The ranked retrieval benchmark (TP2VR) consists of 746 videos from Real-TLV (and 42 queries) and 952 videos (and 70 queries) from AVA.
> >
> > We discuss the datasets and benchmark creation pipeline in more detail in Appendix C. We will be happy to discuss and add any more details that the reviewer thinks are relevant to gauge significance.
> >
> >
> > **W3: The related work is very brief. Lack discussion on weakness of existing papers and fails to build relations between this work and previous works.**
> >
> > To the best of our knowledge, the only other works that explore the use of Temporal Logic for scoring / matching temporal properties over videos are [1, 2]. The key limitations of these approaches include (1) exponential complexity of scoring / matching, rendering them impractical for high-volume tasks such as retrieval, (2) no handling of noisy local predictions (from occlusion, low lighting, etc.), and (3) the use of constant 0.5 threshold for matching.
> >
> > While we discuss the limitations of these methods in the introduction, we have updated the related work section to re-iterate these limitations. We would be happy to cite and discuss any other works that the reviewer considers relevant to ours.
> >
> >
> > **W4: Changes to notation for improved readability**
> >
> > We define the notation and symbols used in Algorithm 1 in Section 2. We have annotated Algorithm 1 with comments on the temporal operators for improved readability. We would also be happy to take suggestions from the reviewer on specific notation that could use clarification.
> >
> > *References*
> >
> > [1] Yang, Y., Gaglione, J. R., Chinchali, S., & Topcu, U. (2023). Specification-driven video search via foundation models and formal verification. arXiv preprint arXiv:2309.10171.
> >
> > [2] Choi, M., Goel, H., Omama, M., Yang, Y., Shah, S., & Chinchali, S. (2024). Towards neuro-symbolic video understanding. In European Conference on Computer Vision (pp. 220-236). Cham: Springer Nature Switzerland.

---

### Official Review · Reviewer_d6dP · 2025-11-01

**Soundness:** 2
**Presentation:** 3
**Contribution:** 2
**Rating:** 4
**Confidence:** 2

**Summary:**

This paper introduces STOPs (Scores for TempOral Properties), aiming to address how to lift potentially noisy, low-level scores from local detectors into global scores for complex temporal properties. To this end, the authors propose a scoring function named LogSTOP, which can efficiently compute these scores. Experiments demonstrate that, on query matching and ranked retrieval tasks, LogSTOP significantly outperforms Large Vision/Audio-Language Models and other baselines.

**Strengths:**

1. The LogSTOP algorithm requires only linear time complexity.
2. The introduced QMTP and TP2VR benchmarks effectively evaluate query matching and ranked retrieval tasks.

**Weaknesses:**

1. The assumption that local properties represent independent events over time is rarely true for real-world sequences and properties. This raises concerns about potential failures in scenarios with complex temporal dependencies. The simplified theoretical basis may therefore not generalize well to timing-dependent cases.

2. LTL cannot express "counting" constraints (e.g., "there are always 2 cars") or handle numeric attributes.

**Questions:**

1. As a key hyperparameter, the smoothing window should be discussed in detail.
2. Is there any exploration of ways to avoid such an assumption?

---

> ### Author Response · Authors · 2025-11-20
> **Response to reviewer d6dP**
>
> We would like to thank the reviewer for their comments. We first answer the reviewer’s questions and then present a discussion on comments mentioned under “weaknesses”.
>
> **Q1: As a key hyperparameter, the smoothing window should be discussed in detail.**
>
> We would like to thank the reviewer for the suggestion and point them to Appendix A for a discussion on the downsampling smoothing window, with examples. In addition to our existing ablation on LogSTOP with and without smoothing in Table 1, we have also added an ablation study of how the value of the smoothing window impacts performance of LogSTOP on query matching in Appendix I.3. Concretely, we consider videos with 20 frames from our QMTP-Video dataset and evaluate LogSTOP (YOLOv8x) with smoothing windows in [1, 2, 4, 5, 10, 20].
>
> *Results:* We find that the balanced accuracies (over matching and non-matching sequences) are highest with w=4 and w=5, and lowest with w=1 and w=20 (please see Table 2 in Appendix I.3). In general, low accuracy with w=1 corroborates our hypothesis of how noisy local predictions can affect the score adversely, while low accuracy with w=20 highlights the loss of information as the sequence is compressed to a single prediction point. Interestingly, we find intuitive trends where increased smoothing benefits or hurts performance depending on the temporal property being matched:
>
> - Properties such as “there is a car in some frame” and “starting at any frame, there is a car in some future frame” match videos where the score for “car” is high in at least one frame. As increased smoothing adversely affects the strength of this single-frame score, the query matching performance improves as the value of w decreases.
> - Properties such as “there is a car and a pedestrian in all frames” and “there is no pedestrian in all frames until there is a car in all frames” match videos where the scores for “car” and “pedestrian” are high in contiguous subsequences of frames. Hence, such properties benefit from increased smoothing, reporting higher query matching accuracies as the value of w increases.
> - For most properties where the two trends can co-occur, moderate values of smoothing are beneficial. For instance, the property “there is no pedestrian in all frames and a car in some frames” matches videos where scores for “pedestrian” are expected to be high in all frames while scores for “car” can be high in a few frames.
>
> **Q2/W1: Is there any exploration of ways to avoid the independence assumption?**
>
> We would like to point the reviewer to the global response R1 for a discussion on alleviating the independence assumptions. To summarize, we could use conditional probabilities over all local properties and timesteps to compute the probability of the temporal property. We could potentially use domain-specific knowledge to compute these conditional probabilities: for instance, given traffic scenarios from overhead cameras, we could compute conditional probabilities under assumptions for vehicle dynamics. Exploring ways to compute these conditional probabilities is an interesting direction for future work.
>
> **W2: LTL cannot express "counting" constraints (e.g., "there are always 2 cars") or handle numeric attributes.**
>
> We would like to argue that the expressiveness of complex temporal properties in LTL is directly related to how the local properties are defined. Hence, the stated property could be expressed if “2 cars” is defined as a local property with an associated predictor. Open vocabulary detection models such as Grounding DINO can provide scores for such local properties and LogSTOP can then be used to score temporal properties over them.
>
> We agree that LTL cannot be used to express this property with “car” as a local property or other properties in regular languages. LogSTOP, however, can easily be extended to properties that can be represented with regular expressions. For instance, the property “there is a car in alternating frames” cannot be expressed in LTL but can be written as the regular expression $(car;(car | \neg car))*$. Since the regular expression can be represented with a DFA, LogSTOP can be extended to score the property with the same computational complexity.

---

> > ### Comment · Reviewer_d6dP · 2025-11-26
> >
> > Thanks for the rebuttal. Most of my concerns have been addressed. I will raise my rating to 6.
> >
> > However, I will not oppose a final decision to reject.

---

> > > ### Author Response · Authors · 2025-12-02
> > > **Thank you**
> > >
> > > We are glad that our response addressed most of your concerns, and we greatly appreciate the increase in your score. Thank you for engaging with us during the rebuttal and for providing constructive feedback that helped strengthen the paper.

---

### Official Review · Reviewer_FdQM · 2025-11-04

**Soundness:** 3
**Presentation:** 4
**Contribution:** 3
**Rating:** 6
**Confidence:** 3

**Summary:**

This paper formalizes the STOP (Assigning Scores for Temporal Properties over sequence) problem and proposes LogSTOP as the efficient scoring function for temporal properties. This paper also proposed two new benchmarks: QMTP and TP2VR for the STOPs problem.

**Strengths:**

1. The notion of using a predictor to score the sequence with local property along with its temporal property is an interesting problem.
2. The proposed scoring function LogSTOP uses dynamic programming and is efficient to compute the score for a sequence.
3. The paper proposed two new benchmarks.

**Weaknesses:**

1. As mentioned in the paper, the relationships for local properties are not completely independent and there are some relations between them. In this case, using probability theory is not guaranteed to be optimal.
2. The threshold of LogSTOP is predefined to be log(0.5) and there is no further analysis. I would like to see the ablation study of that.
3. Although LogSTOP is claimed to be efficient, there is no experiment on the efficiency side, and it will be good if there are comparisons with other models.
4. The models used for comparison are limited (only 2 for each dataset and there are only 7B models’ results), which does not give a holistic evaluation of the effectiveness of the proposed method.

**Questions:**

1. In figure 4, why does the accuracy of LogSTOP decrease after using the log(0.5) threshold? Does it mean it include the  log(0.5) threshold mentioned before or it means a hard log(0.5) threshold?
2. Have authors tested the mixed modality case for the STOP problem?

---

> ### Author Response · Authors · 2025-11-20
> **Response to reviewer FdQM**
>
> We appreciate the reviewer’s positive remarks regarding the problem definition and the efficiency of LogSTOP. We would first like to answer the questions before presenting results from new experiments that address the questions/comments mentioned under “weaknesses”.
>
> **Q1: In figure 4, why does the accuracy of LogSTOP decrease after using the log(0.5) threshold?**
>
> We assume that the reviewer is referring to log(0.5) in Table 1. The log(0.5) threshold in Table 1 refers to the hard / constant log(0.5) threshold. As we demonstrate in Figure 3, this reduction in accuracy can be attributed to the constant log(0.5) threshold rejecting more matching sequences than the adaptive log(0.5) threshold.
>
> **Q2: Have authors tested the mixed modality case for the STOP problem?**
>
> We do not have experiments with multi-modal datasets. Our benchmark creation pipeline requires annotations for local properties (i.e., frame/segment level annotations for objects/actions/emotions/etc.) for computing the ground truth temporal labels and we are not aware of any multimodal datasets that provide such annotations. We would be happy to take dataset suggestions from the reviewer.
>
> **W1: Optimality of probability theory**
>
>  We would like to point the reviewer to our detailed comments on the probabilistic interpretation of LogSTOP in the global response (R1). To summarize, we do not want to claim that LogSTOP computes the true probability of temporal properties. Our goal is instead to propose an efficient scoring function for temporal properties that is useful for certain downstream applications, and we find that using probability theory for aggregating local scores performs well in practice on a variety of applications and domains.
>
> **W2: Ablation study on the log(0.5) threshold**
>
> We would like to thank the reviewer for this suggestion. Our choice of log 0.5 as the base threshold was motivated by prior work [1,2]. We have now included an ablation with a sweep over base threshold values in Appendix I.2. Concretely, we focus on videos with 50 frames (the longest frame range) from QMTP-Video and compute LogSTOP’s query matching accuracy with 10 values of base thresholds (log 0.1, log 0.2, …, log 0.9). We find that LogSTOP reports high balanced accuracies on query matching with base thresholds around log 0.5 and the accuracy decreases as we move further away from log 0.5 (Figure 7 in Appendix I.2). Interestingly, the range of good thresholds varies with the object detection model used. LogSTOP with YOLOv8 reports good matching performance with a wider range of thresholds than with Grounding DINO.
>
> **W3: Efficiency plots for LogSTOP**
>
> We have added efficiency plots for LogSTOP in Appendix I.4. Concretely, for each temporal property template, we plot the average time taken to compute LogSTOPs for sequences of different lengths (Figure 8 in Appendix I.4). We find a linear trend – the time taken scales roughly linearly with the length of the sequence for a given temporal property – which corroborates our computational complexity analysis in Section 3.
>
> **W4: Other models**
>
> We have included results with a larger LVLM, InternVL2-26B, on the query matching task. The results are presented in Table 7 in Appendix J.2. With 5 A100 GPUs, the model can only support videos with up to 30 frames and hence, we restrict comparison with other methods on this subset of QMTP-Video. LogSTOP with Grounding DINO and YOLOv8 outperforms this model by 3% and 13% in terms of balanced accuracy. Interestingly, the model also performs only marginally better than LongVA-7B (1% improvement in accuracy) despite being much larger.
>
> We would like to emphasize all LVLM and LALM models used for comparison are significantly larger than the ones used as local property predictors by LogSTOP (YOLOv8x, Grounding DINO, HuBERT, and SlowR50 have 90M, 172M, 1B, 34.57M parameters respectively).
>
> These new results with InternVL2-26B further support our existing findings on the limitations of LVLMs on temporal reasoning tasks and long video sequences. We would also be happy to hear the reviewer’s suggestions on other open-source models that should be included (and can ideally support up to 50 video frames as input).
>
> *References*
>
> [1] Yang, Y., Gaglione, J. R., Chinchali, S., & Topcu, U. (2023). Specification-driven video search via foundation models and formal verification. arXiv preprint arXiv:2309.10171.
>
> [2] Choi, M., Goel, H., Omama, M., Yang, Y., Shah, S., & Chinchali, S. (2024). Towards neuro-symbolic video understanding. In European Conference on Computer Vision (pp. 220-236). Cham: Springer Nature Switzerland.

---

### Author Response · Authors · 2025-11-20
**Global response**

We would like to thank the reviewers for their insightful comments and suggestions. In this global response, we would like to discuss concerns raised by multiple reviewers and highlight the key updates to the paper.

**R1. LogSTOP is an efficient scoring function that does not represent the true probability of temporal properties**

Our main aim with this work is to formalize the problem of assigning scores to temporal properties and to identify an efficient method for computing scores that are useful for several downstream applications. As we discuss in the paper and as the reviewers note, the independence assumptions prohibit us from treating LogSTOPs as true probabilities. Instead, we draw on ideas from probability theory purely as tools for aggregation. This choice allows LogSTOP to operate efficiently while empirically outperforming various baselines (including alternative aggregation methods) on applications such as query matching and ranked retrieval.

We would also like to discuss potential directions for relaxing the independence assumptions using conditional probabilities of local properties. For instance, consider the temporal property
“There is a car and a person in all frames”. Using chain rule, the probability of this property can be written as:

$$
P(car_0 \land person_0 \land \dots \land car_T \land person_T)
= P(car_0)
  \cdot P(person_0 \mid car_0)
  \cdot P(car_1 \mid car_0, person_0)
  \cdot P(person_1 \mid car_0, car_1, person_0)
  \cdots
$$

where  $car\_i$  refers to a car being present in frame $i$.

This decomposition significantly increases the difficulty of the problem: accurate conditional probabilities are difficult to obtain and using them to compute probabilities of temporal properties is likely computationally expensive. We believe that these challenges point to promising avenues for future research – using domain-specific knowledge as priors for computing the conditional probabilities is a possible direction.

**R2. Supporting ablations and analysis**

We would like to thank the reviewers for suggestions on additional ablations and analyses. We have added the following ablations and supporting experiments to the paper:

- In Appendix I.3, we present an ablation study on the effect of the smoothing window (w) on LogSTOP’s query matching performance. We find that moderate values of w report the highest balanced accuracies and there exist intuitive trends for different temporal properties in terms of whether they benefit from increased/decreased smoothing (Table 2).
- In Appendix I.2, we present an ablation study on how changing the base threshold value from log 0.5 to other values in [log 0.1, …, log 0.9] affects LogSTOP’s query matching performance. We find that the highest accuracies are reported using log 0.5 as the base threshold value and the accuracy decreases as we move further away from log 0.5 (Figure 7).
- In Appendix I.4, we present efficiency plots for LogSTOP demonstrating that the time taken to compute LogSTOPs scales linearly with the length of the sequence for any given temporal property as expected (Figure 8).
- In Appendix J.2, we report query matching results with a much larger LVLM, namely InternVL2-26B, on QMTP-Video. We find that this model performs only marginally better than LongVA-7B (1% improvement) and LogSTOP with YOLO outperforms this model by 13% in terms of balanced accuracy (Table 7).

We discuss the setup and results for these experiments in more detail in the responses to respective reviewers.

---

### Author Response · Authors · 2025-12-02
**Overview of key contributions and new experiments added during the rebuttal**

We thank the reviewers for their time, for recognizing the novelty of our contributions, and for providing valuable suggestions for additional experiments. In response, we have incorporated new experiments and analyses, which we believe have further strengthened the paper. Below, we first summarize the key contributions of our work, followed by a list of the new supporting experiments conducted based on the reviewers’ feedback:

### **Key contributions**

1. We formalize the problem of assigning scores to complex temporal properties over sequences. As highlighted by reviewers E262, FdQM, and 4Lc6, *our formulation is clear and captures an interesting, intuitive, and challenging problem*.
2. We present LogSTOP, an algorithm that assigns scores to Linear Temporal Logic properties in time linear in the lengths of both the sequence and the property. Reviewers FdQM, d6dP, and E262 noted that *LogSTOP is efficient and scalable, making it suitable for practical applications such as large-scale retrieval*. Moreover, as emphasized by reviewer E262 and demonstrated in our experiments, *LogSTOP is explicitly designed to handle noisy real-world data and performs effectively in practice*.
3. We introduce two benchmarks for downstream tasks: Query Matching with Temporal Properties (QMTP) and Temporal Property to Video Retrieval (TP2VR). These benchmarks span video and speech modalities and include local properties such as objects and actions in videos and emotions in speech. Reviewers FdQM, 4Lc6, and d6dP highlighted that *the benchmarks are novel and provide effective evaluations for query matching and ranked retrieval, making them valuable to the ICLR community*.
4. For query matching, we propose a length and query-adaptive threshold for LogSTOP. Using YOLOv8x and HuBERT as local property predictors for objects in videos and emotions in speech, respectively, LogSTOP outperforms Large Vision and Audio Language Models as well as an exponential-time baseline on the QMTP benchmark by *more than 16% in terms of accuracy*.
5. For ranked retrieval, we propose a quadratic-time algorithm that leverages LogSTOP to rank sequences by their maximum subsequence scores. With YOLOv8x and SlowR50 as local property predictors for objects and actions in videos, respectively, LogSTOP outperforms zero-shot text-to-video retrieval methods by *more than 19% in mean average precision and 16% in recall*.


### **New supporting experiments added during the rebuttal**

1. **Effect of smoothing window on LogSTOP:** In Appendix I.3, we present an ablation study on the effect of the smoothing window (w) on LogSTOP’s query matching performance. We find that moderate values of w report the highest balanced accuracies and there exist intuitive trends for different temporal properties in terms of whether they benefit from increased/decreased smoothing (Table 2). We discuss these results in more detail in response to Q1 from reviewer d6dP.

2. **Effect of base threshold value on query matching performance:** In Appendix I.2, we present an ablation study on how changing the base threshold value from log 0.5 to other values in [log 0.1, …, log 0.9] affects LogSTOP’s query matching performance. We find that the highest accuracies are reported using log 0.5 as the base threshold value and the accuracy decreases as we move further away from log 0.5 (Figure 7).

3. **Efficiency plots for LogSTOP:** In Appendix I.4, we present efficiency plots for LogSTOP demonstrating that the time taken to compute LogSTOPs scales linearly with the length of the sequence for any given temporal property as expected (Figure 8).

4. **Comparison with larger VLMs:** In Appendix J.2, we report query matching results with a much larger VLM, namely InternVL2-26B, on QMTP-Video. We find that this model performs only marginally better than LongVA-7B (1% improvement). Moreover, LogSTOP with YOLOv8x (90M parameters) outperforms this model by 13% in terms of balanced accuracy (Table 7). We discuss these results in more detail in response to W4 from reviewer FdQM.

---

### Meta-Review · Area_Chair_iHr4 · 2025-12-05

**Summary:**

This paper studies how to lift local property detection scores in videos or audio to sequence-level temporal property scores. The authors formalize this task as the “Scoring for TempOral Properties (STOPs)” problem, aiming to map these scores to temporal properties expressed in Linear Temporal Logic (LTL) despite potential noise in local predictions. To address this, the paper proposes a scoring function, LogSTOP, which efficiently computes sequence-level temporal property scores and handles temporal patterns such as “eventually” or “until”. This paper also proposed two new benchmarks: QMTP and TP2VR for the STOPs problem. The experiments show some good performance.

All reviewers raise concerns about the choice of probability theory. The proposed method relies on a strong independence assumption: the relationships for local properties are not completely independent, and there are some relations between them; therefore, it is rarely true for real-world sequences and properties. Moreover, the experiments are insufficient, lacking generalization evaluations on video and speech domains for investigating their domain-agnostic ability.

Based on the above concerns, the AC recommends Reject. The authors are encouraged to revise their works for the next cycle.

**Reviewer Concerns:**

Concerns were addressed:
1. Ablation on the threshold of LogSTOP.
2. Efficiency analysis.
3. More experimental comparison.
4. Implementation details, such as the smoothing window.

Concerns remain:
1. The proposed method relies on a strong independence assumption: the relationships for local properties are not completely independent, and there are some relations between them; therefore, it is rarely true for real-world sequences and properties.
2. The experiments are insufficient, lacking generalization evaluations on video and speech domains for investigating their domain-agnostic ability.

**Reviewer Scores:**

Since the core concerns of all reviewers are not addressed, the discussion may not influence their ratings.

---

### Decision · Program_Chairs · 2026-01-26

Reject